# A bipartite iron-dependent transcriptional regulation of the tryptophan salvage pathway in *Chlamydia trachomatis*

Nick D Pokorzynski, Amanda J Brinkworth, Rey Carabeo*

Center for Reproductive Biology, School of Molecular Biosciences, College of Veterinary Medicine, Washington State University, Pullman, United States

**Abstract** During infection, pathogens are starved of essential nutrients such as iron and tryptophan by host immune effectors. Without conserved global stress response regulators, how the obligate intracellular bacterium *Chlamydia trachomatis* arrives at a physiologically similar 'persistent' state in response to starvation of either nutrient remains unclear. Here, we report on the iron-dependent regulation of the *trpRBA* tryptophan salvage pathway in *C. trachomatis*. Iron starvation specifically induces *trpBA* expression from a novel promoter element within an intergenic region flanked by *trpR* and *trpB*. YtgR, the only known iron-dependent regulator in *Chlamydia*, can bind to the *trpRBA* intergenic region upstream of the alternative *trpBA* promoter to repress transcription. Simultaneously, YtgR binding promotes the termination of transcripts from the primary promoter upstream of *trpR*. This is the first description of an iron-dependent mechanism regulating prokaryotic tryptophan biosynthesis that may indicate the existence of novel approaches to gene regulation and stress response in *Chlamydia*.
DOI: https://doi.org/10.7554/eLife.42295.001

## Introduction

Nutrient acquisition is critical for the success of pathogenic bacteria. Many pathogenic bacteria must siphon nutrients from their hosts, such as nucleotides, amino acids and biometals (*Brown et al., 2008*; *Eisenreich et al., 2010*; *Ray et al., 2009*; *Skaar, 2010*). This common feature among pathogens renders them susceptible to nutrient limitation strategies associated with the host immune response (*Hood and Skaar, 2012*). Counteractively, bacterial pathogens have evolved sophisticated molecular mechanisms to respond to nutrient deprivation, involving increasingly complex and sophisticated nutrient-sensing regulatory networks. These stress response mechanisms are essential for pathogens to avoid clearance by the immune system. By delineating their function at the molecular level, we can better target aspects of the host-pathogen interface suitable for therapeutic manipulation. However, stress responses in the obligate intracellular bacterium *Chlamydia trachomatis* are relatively poorly characterized, leaving unanswered many fundamental questions about the biology of this pathogen.

*C. trachomatis* is the leading cause of bacterial sexually transmitted infections (STIs) and infection-derived preventable blindness worldwide (*CDC, 2017*; *Newman et al., 2015*; *Taylor et al., 2014*). Genital infections of chlamydia disproportionately affect women and are associated with serious sequelae in the female reproductive tract such as tubal factor infertility (*Hafner, 2015*). Chlamydiae are Gram-negative bacterial parasites that develop within a pathogen-specified membrane-bound organelle termed the inclusion (*Moore and Ouellette, 2014*). Chlamydial development is uniquely characterized by a biphasic interconversion of an infectious elementary body (EB) with a non-infectious, but replicative reticulate body (RB) (*Abdelrahman and Belland, 2005*). An obligate intracellular lifestyle has led to reductive genome evolution across chlamydial species; Chlamydiae

*For correspondence: rey.carabeo@wsu.edu

Competing interests: The authors declare that no competing interests exist.

**eLife digest** All forms of life must take up nutrients from their environment to survive. *Chlamydia trachomatis*, a bacterium that causes many sexually-transmitted infections, is no exception. These bacteria do not normally make one of the building blocks of proteins, the amino acid tryptophan, but instead scavenge it from their human host.

One way that the immune system tries to fight a chlamydia infection is by cutting off the supply of tryptophan in an attempt to starve the bacteria. But the microbes have evolved to respond to these hardships and keep themselves alive. The 'tryptophan salvage pathway' is a set of genes that, when switched on, allows the *Chlamydia* bacteria to take up a molecule found in the female genital tract that they can use to make their own tryptophan. Yet, how do the bacteria know when to activate these genes?

Tryptophan starvation is not the only strategy that the immune system uses to fight chlamydia. It also restricts the supply of the essential metal iron to these bacteria. Now, using human cells grown in the laboratory and infected with *Chlamydia* bacteria, Pokorzynski et al. show that iron starvation switches on the tryptophan salvage pathway. *Chlamydia* most likely senses changes in iron levels via a protein called YtgR, and a closer look at the bacterial DNA revealed that YtgR interacts with the genes of the tryptophan salvage pathway. When iron levels were high, YtgR locked on to the DNA in the middle of this set of genes. This effectively switched off the genes on either side of the binding site. When iron levels dropped, YtgR came away from the DNA, releasing the genes and allowing the cell to use them to start making its own tryptophan. Together these findings indicate that, when the bacteria sense that iron levels have dropped, they prepare for a shortage of tryptophan too.

Chlamydia is the most common bacterial sexually transmitted infection worldwide. Left untreated, it can cause infertility and blindness. This and future studies aimed at understanding how these bacteria respond to immune attack may reveal new ways to prevent or treat these infections.
DOI: https://doi.org/10.7554/eLife.42295.002

have retained genes uniquely required for their survival, but have become nutritionally dependent on their hosts by discarding many metabolism-related genes (*Clarke, 2011*). Of note, *C. trachomatis* does not possess genes necessary for eliciting a stringent response to nutrient starvation (*e.g. relA, spoT*), suggesting that this pathogen may utilize novel mechanisms to respond to nutrient stress (*Stephens et al., 1998*).

It is well established that in response to various stressors, Chlamydiae deviate from their normal developmental program to initiate an aberrant developmental state, termed 'persistence' (*Wyrick, 2010*). This persistent state is distinguished by the presence of viable, but non-cultivable, abnormally enlarged chlamydial organisms that display dysregulated gene expression. Importantly, *Chlamydia* can be reactivated from persistence by abatement of the stress condition. As such, chlamydial persistence at least superficially resembles a global stress response mechanism. Yet the mechanistic underpinnings of this phenotype are poorly understood, with most published studies focusing on the molecular and metabolic character of the aberrant, persistent form. It is therefore unclear to what extent primary stress responses contribute to the global persistent phenotype in *Chlamydia*.

The best described inducer of persistence is the pro-inflammatory cytokine interferon-gamma (IFN-γ). The bacteriostatic effect of IFN-γ has been primarily attributed to host cell tryptophan (Trp) catabolism, an amino acid for which *C. trachomatis* is auxotrophic (*Byrne et al., 1986*; *Fehlner-Gardiner et al., 2002*; *Taylor and Feng, 1991*). Following IFN-γ stimulation, infected host cells up-regulate expression of indoleamine-2,3-dioxygenase (IDO1), which catabolizes Trp to *N*-formylkynurenine via cleavage of the indole ring (*Macchiarulo et al., 2009*). *C. trachomatis* cannot recycle kynurenines, unlike some other chlamydial species (*Wood et al., 2004*), and thus IFN-γ stimulation effectively results in Trp starvation to *C. trachomatis*. The primary regulatory response to Trp starvation in *C. trachomatis* is mediated by a TrpR ortholog, whose Trp-dependent binding to cognate promoter elements represses transcription (*Akers and Tan, 2006*; *Carlson et al., 2006*). This mechanism of regulatory control is presumably limited in *C. trachomatis*, as homologs of genes regulated

by TrpR in other bacteria (*e.g. trpF, aroH, aroL*) have not been shown to respond to Trp limitation (*Wood et al., 2003*).

In many Gram-negative bacteria, such as *Escherichia coli*, *trpR* is monocistronic and distal to the Trp biosynthetic operon. In *C. trachomatis*, TrpR is encoded in an operon, *trpRBA*, which also contains the Trp synthase α- and β- subunits (TrpA and TrpB, respectively), and possesses a 348 base-pair (bp) intergenic region (IGR) that separates *trpR* from *trpBA*. TrpBA catalyzes the final steps of Trp biosynthesis in bacteria; TrpA converts indoleglycerol-3-phosphate (IGP) to indole which is then condensed with serine by TrpB to form Trp. In *C. trachomatis*, TrpA cannot bind IGP and thus *C. trachomatis* requires indole as a substrate to synthesize Trp (*Fehlner-Gardiner et al., 2002*). Despite significant research on the chlamydial *trpRBA* operon, the functional significance of the *trpRBA* IGR is poorly characterized. While a putative TrpR operator sequence was identified in the IGR overlapping an alternative transcriptional origin for *trpBA* (*Carlson et al., 2006*), TrpR binding was not observed (*Akers and Tan, 2006*). Based on in silico predictions, an attenuator sequence has been annotated within the *trpRBA* IGR (*Merino and Yanofsky, 2005*), but this has not been thoroughly validated experimentally. Regardless, the IGR is >99% conserved at the nucleotide sequence level across ocular, genital and lymphogranuloma venereum (LGV) serovars of *C. trachomatis*, indicating functional importance (*Carlson et al., 2005*; *Seth-Smith et al., 2009*; *Stephens et al., 1998*; *Thomson et al., 2008*). Therefore, outside of TrpR-mediated repression, the complete detail of *trpRBA* regulation remains poorly elucidated and previous reports have indicated the possibility of more complex mechanisms of regulation (*Brinkworth et al., 2018*).

The requirement of TrpBA for *C. trachomatis* to survive IFN-γ-mediated Trp starvation is well documented. However, IFN-γ is also known to limit iron and other essential biometals to intracellular pathogens as a component of host nutritional immunity (*Cassat and Skaar, 2013*; *Hood and Skaar, 2012*). Whether *C. trachomatis* has adapted to respond to these various IFN-γ-mediated insults remains unclear. *Chlamydia* have a strict iron dependence for normal development, evidenced by the onset of persistence following prolonged iron limitation (*Raulston, 1997*). Importantly, *Chlamydia* presumably acquire iron via vesicular interactions between the chlamydial inclusion and slow-recycling transferrin (Tf)-containing endosomes (*Ouellette and Carabeo, 2010*). IFN-γ is known to down-regulate transferrin receptor (TfR) expression in both monocytes and epithelial cells with replicative consequences for resident intracellular bacteria (*Byrd and Horwitz, 1993*; *Byrd and Horwitz, 1989*; *Igietseme et al., 1998*; *Nairz et al., 2008*). However, iron homeostasis in *Chlamydia* is poorly understood due to the lack of functionally characterized homologs to iron acquisition machinery that are highly conserved in other bacteria (*Pokorzynski et al., 2017*). Only the *ytgABCD* operon, encoding a metal permease, has been clearly linked to iron acquisition (*Miller et al., 2009*). Intriguingly, the YtgC (CTL0325) open reading frame (ORF) encodes a N-terminal permease domain fused to a C-terminal DtxR-like repressor domain, annotated YtgR (*Akers et al., 2011*; *Thompson et al., 2012*). YtgR is cleaved from the permease domain during infection and functions as an iron-dependent transcriptional repressor to autoregulate the expression of its own operon (*Thompson et al., 2012*). YtgR represents the only identified iron-dependent transcriptional regulator in *Chlamydia*. Whether YtgR maintains a more diverse transcriptional regulon beyond the *ytgABCD* operon has not yet been addressed and remains an intriguing question in the context of immune-mediated iron limitation to *C. trachomatis*.

Consistent with the highly reduced capacity of the chlamydial genome, it is likely that *C. trachomatis* has a limited ability to tailor a specific response to each individual stress. In the absence of identifiable homologs for most global stress response regulators in *C. trachomatis*, we hypothesized that primary stress responses to pleiotropic insults may involve mechanisms of co-regulation by stress-responsive transcription factors. Here, we report on the unique iron-dependent regulation of the *trpRBA* operon in *Chlamydia trachomatis*. We propose a model of iron-dependent transcriptional regulation of *trpRBA* mediated by the repressor YtgR binding specifically to the IGR, which would enable *C. trachomatis* to respond similarly to the antimicrobial deprivation of Trp or iron mediated by IFN-γ. Such a mechanism of iron-dependent regulation of Trp biosynthesis has not been previously described in any other prokaryote and adds to the catalog of regulatory models for Trp biosynthetic operons in bacteria. Further, it reveals a highly dynamic mode of regulatory integration within the *trpRBA* operon, employing bipartite control at the transcription initiation and termination steps.

## Results

### Brief iron limitation via 2,2-bipyridyl treatment yields iron-starved, but non-persistent *Chlamydia trachomatis*

To identify possible instances of regulatory integration between iron and Trp starvation in *C. trachomatis*, we optimized a stress response condition that preceded the development of a characteristically persistent phenotype. We reasoned that in order to effectively identify regulatory integration, we would need to investigate the bacterium under stressed, but not aberrant, growth conditions such that we could distinguish primary stress responses from abnormal growth. To specifically investigate the possible contribution of iron limitation to a broader immunological (*e.g.* IFN-γ-mediated) stress, we utilized the membrane-permeable iron chelator 2,2-bipyridyl (Bpdl), which has the advantage of rapidly and homogeneously starving *C. trachomatis* of iron (*Thompson and Carabeo, 2011*). We chose to starve *C. trachomatis* serovar L2 of iron starting at 12 hr post-infection (hpi), or roughly at the beginning of mid-cycle growth. At this point the chlamydial organisms represent a uniform population of replicative RBs that are fully competent, both transcriptionally and translationally, to respond to stress. We treated infected HeLa cell cultures with 100 µM Bpdl or mock for either 6 or 12 hr (hrs) to determine a condition sufficient to limit iron to *C. trachomatis* without inducing hallmark persistent phenotypes. We stained infected cells seeded on glass coverslips with convalescent human sera and analyzed chlamydial inclusion morphology under both Bpdl- and mock-treated conditions by laser point-scanning confocal microscopy (*Figure 1A*). Following 6 hr of Bpdl treatment, chlamydial inclusions were largely indistinguishable from mock-treated inclusions, containing a homogeneous population of larger organisms, consistent with RBs in mid-cycle growth. However, by 12 hr of Bpdl treatment, the inclusions began to display signs of aberrant growth: they were perceptibly smaller, more comparable in size to 18 hpi, and contained noticeably fewer organisms, perhaps indicating a defect in bacterial replication or RB-to-EB differentiation. These observations were consistent with our subsequent analysis of genome replication by quantitative PCR (qPCR; *Figure 1B*). At 6 hr of Bpdl treatment, there was no statistically distinguishable difference in genome copy number when compared to the equivalent mock-treated time-point. However, by 12 hr of treatment, genome copy number was significantly reduced 4.47-fold in the Bpdl-treated group relative to mock-treatment (p=0.00151). We then assayed the transcript expression of two markers for persistence by reverse transcription quantitative PCR (RT-qPCR): the early gene *euo*, encoding a transcriptional repressor of late-cycle genes (*Figure 1C*), and the adhesin *omcB*, which is expressed late in the developmental cycle (*Figure 1D*). Characteristic persistence would display elevated *euo* expression late into infection and suppressed *omcB* expression throughout development. We observed that at 6 hr of Bpdl treatment, there was no statistically distinguishable difference in either *euo* or *omcB* expression when compared to the mock-treatment. Still at 12 hr of Bpdl treatment, *euo* expression was unchanged. However, *omcB* expression was significantly up-regulated following 12 hr of Bpdl-treatment (p=0.0024). This was unexpected, but we note that *omcB* expression has been shown to vary between chlamydial serovars and species when starved for iron (*Pokorzynski et al., 2017*). The decision to begin our brief iron starvation at 12 hpi may produce notable transcriptional differences from previous studies in which iron starvation was induced at the beginning of the chlamydial developmental cycle, and thereby prevented the establishment of a normal transcriptional program by *Chlamydia*. Collectively, these data indicated that 6 hr of Bpdl treatment was a more suitable time-point at which to monitor iron-limited stress responses.

We additionally assayed these same metrics following 6 or 12 hr of Trp starvation by culturing cells in either Trp-replete or Trp-depleted DMEM-F12 media supplemented with fetal bovine serum (FBS) pre-dialyzed to remove amino acids. We observed no discernable change in inclusion morphology out to 12 hr of Trp starvation (*Figure 1—figure supplement 1A*), but genome copy numbers were significantly reduced 2.7-fold at this time-point (p=0.00612; *Figure 1—figure supplement 1B*). The transcript expression of *euo* (*Figure 1—figure supplement 1C*) and *omcB* (*Figure 1—figure supplement 1D*) did not significantly change at either treatment duration, but Trp-depletion did result in a 1.88-fold reduction in *omcB* expression (p=0.0544), consistent with a more characteristic persistent phenotype. These data therefore also indicated that 6 hr of treatment would be ideal to monitor non-persistent responses to Trp limitation.

We next sought to determine whether our brief 6 hr Bpdl treatment was sufficient to elicit a transcriptional iron starvation phenotype. We chose to analyze the expression of three previously

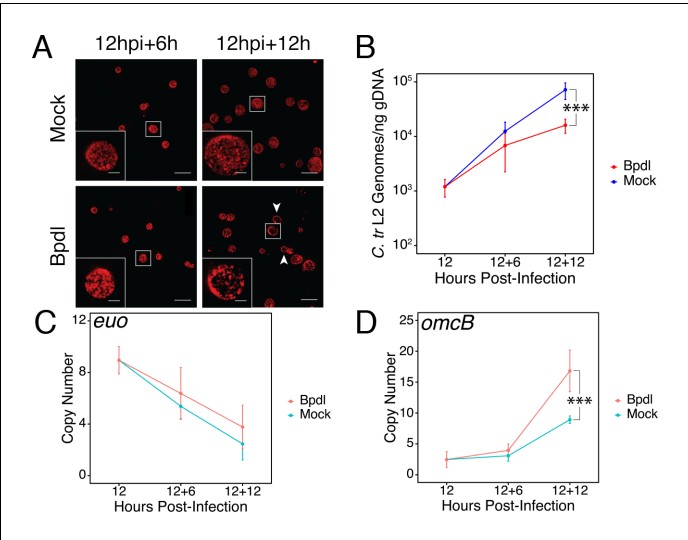

**Figure 1.** Brief iron limitation via 2,2-bipyridyl treatment precedes the onset of characteristic chlamydial persistence. (A) *C. trachomatis* L2-infected HeLa cells were fixed and stained with convalescent human sera to image inclusion morphology by confocal microscopy following Bpdl treatment at the indicated times post-infection. Arrowheads indicate inclusions with visibly fewer organisms in the 12 hr Bpdl-treated condition. Figure shows representative experiment of three biological replicates. Scale bar = 25 μm, Inset scale bar = 5 μm. (B) Genomic DNA (gDNA) was harvested from infected HeLa cells at the indicated times post-infection under iron-replete (blue) and -depleted (red) conditions. Chlamydial genome copy number was quantified by qPCR. Chlamydial genome replication is stalled following 12 hr of Bpdl treatment, but not 6. N = 3. (C) Total RNA was harvested from infected HeLa cells at the indicated times post-infection under iron-replete (teal) and -depleted (orange) conditions. The transcript abundance of hallmark persistence genes *euo* and (D) *omcB* were quantified by RT-qPCR and normalized against genome copy number. Only at 12 hr of Bpdl treatment is *omcB* expression significantly affected. N = 3. Statistical significance was determined by One-Way ANOVA followed by post-hoc pairwise *t*-tests with Bonferroni's correction for multiple comparisons. *=$p < 0.05$, **=$p < 0.01$, ***=$p < 0.005$.
DOI: https://doi.org/10.7554/eLife.42295.003

The following source data and figure supplement are available for figure 1:

**Source data 1.** Source data for *Figure 1A–D*, *Figure 1—figure supplement 1*.
DOI: https://doi.org/10.7554/eLife.42295.005

**Figure supplement 1.** Brief media-defined tryptophan limitation does not produce characteristically persistent *Chlamydia*.
DOI: https://doi.org/10.7554/eLife.42295.004

identified iron-regulated transcripts, *ytgA* (*Figure 2A*), *ahpC* (*Figure 2B*) and *devB* (*Figure 2C*), by RT-qPCR under Bpdl- and mock-treated conditions (*Dill et al., 2009*; *Thompson and Carabeo, 2011*). In addition, we analyzed the expression of one non-iron-regulated transcript, *dnaB* (*Figure 2D*), as a negative control (*Brinkworth et al., 2018*). Following 6 hr of Bpdl treatment, we observed that the transcript expression of the periplasmic iron-binding protein *ytgA* was significantly elevated 1.75-fold relative to the equivalent mock-treated time-point (p=0.0052). However, we did not observe induction of *ytgA* transcript expression relative to the 12 hpi time-point. Here, we considered that apparent increases in transcription could be due to two factors: developmental regulation and transcriptional response to stress. Therefore, expression of genes of interest were monitored over time, for example 18 versus 12 hpi, in addition to single-timepoint comparisons, for example 18 hpi only. While we did not observe induction of *ytgA* over time, which would be more consistent with an iron-starved phenotype (*i.e.* 'turning on' gene expression), we reason that this is a consequence of the brief treatment period. This is in agreement with the need to prolong iron chelation to observe the transcriptional induction of *ytgA* (*Miller et al., 2009*; *Raulston et al., 2007*; *Thompson and Carabeo, 2011*). Similarly, we observed that the transcript expression of the thioredoxin *ahpC* was significantly elevated 2.15-fold relative to the equivalent mock-treated time-point (p=0.038) but was not induced relative to the 12 hpi time-point. The modestly elevated

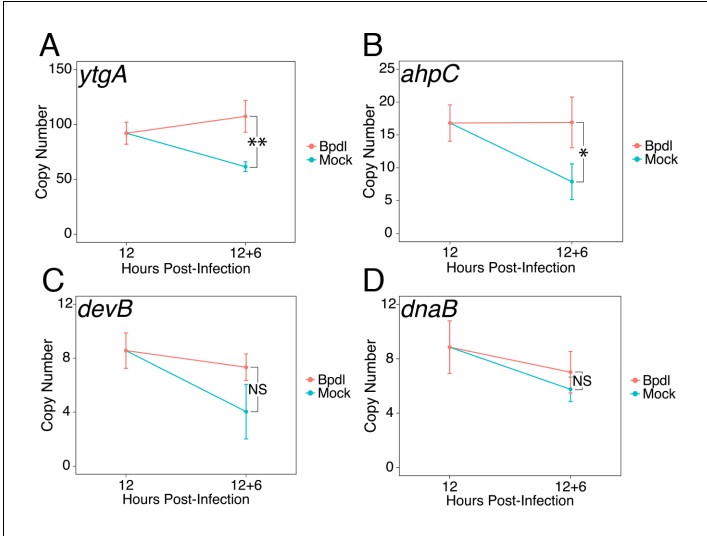

**Figure 2.** Brief iron limitation condition produces mild iron-starved transcriptional phenotype. (**A**) Total RNA and gDNA was harvested from infected HeLa cells at the indicated times post-infection under iron-replete (teal) and -depleted (orange) conditions. The transcript abundance of iron-regulated *ytgA*, (**B**) *ahpC*, (**C**) *devB* and (**D**) non-iron regulated *dnaB* were quantified by RT-qPCR and normalized against genome copy number. The transcript expression of *ytgA* and *ahpC* were significantly elevated following 6 hr BpdI treatment, indicative of iron starvation to *C. trachomatis*. N = 3. Statistical significance was determined by One-Way ANOVA followed by post-hoc pairwise *t*-tests with Bonferroni's correction for multiple comparisons. *=$p < 0.05$, **=$p < 0.01$, ***=$p < 0.005$.
DOI: https://doi.org/10.7554/eLife.42295.006

The following source data is available for figure 2:

**Source data 1.** Source data for *Figure 2A–D*.
DOI: https://doi.org/10.7554/eLife.42295.007

expression of these genes likely represents *bona fide* transcriptional responses to iron starvation given that the treatment condition was optimized to avoid gross changes in chlamydial development. The transcript expression of *devB*, encoding a 6-phosphogluconolactonase involved in the pentose phosphate pathway, was not observed to significantly respond to our brief iron limitation condition, suggesting that it is not a component of the primary iron starvation stress response in *C. trachomatis*. As expected, the transcript expression of *dnaB*, a replicative DNA helicase, was not altered by our iron starvation condition, consistent with its presumably iron-independent regulation (*Brinkworth et al., 2018*). Overall, these data confirmed that our 6 hr BpdI treatment condition was suitable to produce a mild iron starvation phenotype at the transcriptional level, facilitating our investigation of iron-dependent regulatory integration.

## Transcript expression of the *trpRBA* operon is differentially regulated by iron in *Chlamydia trachomatis*

Upon identifying an iron limitation condition that produced a relevant transcriptional phenotype while avoiding the onset of persistent development, we aimed to investigate whether the immediate response to iron starvation in *C. trachomatis* would result in the consistent induction of pathways unrelated to iron utilization/acquisition, but nevertheless important for surviving immunological stress. The truncated Trp biosynthetic operon, *trpRBA* (*Figure 3A*), has been repeatedly linked to the ability of genital and LGV serovars (D-K and L1-3, respectively) of *C. trachomatis* to counter IFN-γ-mediated stress. This is due to the capacity of the chlamydial Trp synthase in these serovars to catalyze the β synthase reaction, that is the condensation of indole to the amino acid serine to form Trp (*Fehlner-Gardiner et al., 2002*). In the presence of exogenous indole, *C. trachomatis* is therefore able to biosynthesize Trp such that it can prevent the development of IFN-γ-mediated persistence. Correspondingly, the expression of *trpRBA* is highly induced following IFN-γ stimulation of infected cells (*Belland et al., 2003*; *Østergaard et al., 2016*). These data have historically implicated Trp

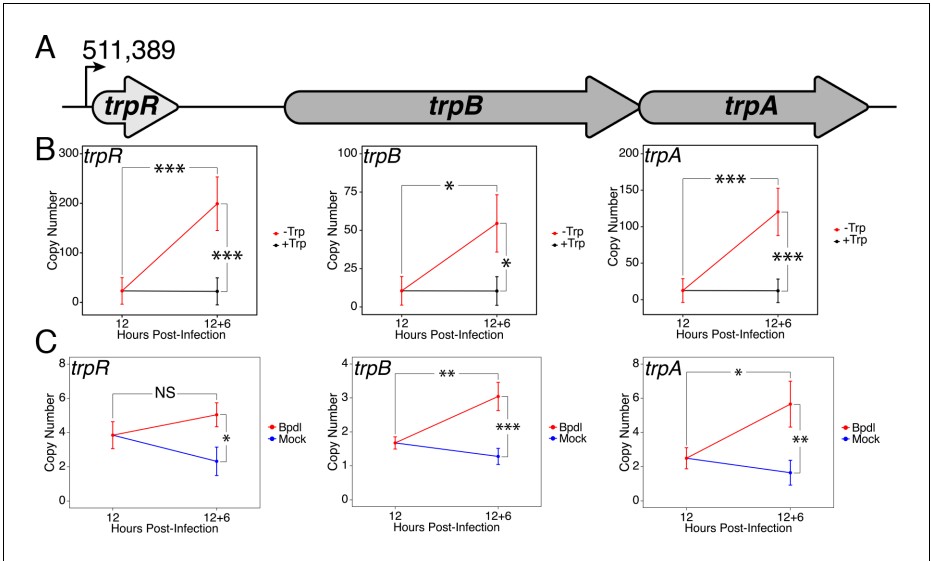

**Figure 3.** Expression of the *trpRBA* operon in *C. trachomatis* is differentially regulated by brief iron limitation. (**A**) Cartoon depiction of the *trpRBA* operon (drawn to scale) with the primary transcriptional start site upstream of *trpR* annotated. (**B**) Total RNA and gDNA were harvested from infected HeLa cells at the indicated times post-infection under Trp-replete (black) and -depleted (red) conditions. The transcript expression of *trpRBA* operon was quantified by RT-qPCR and normalized against genome copy number. All three ORFs are significantly induced relative to 12 hpi following Trp starvation. N = 3. (**C**) Total RNA and gDNA were harvested from infected HeLa cells at the indicated times post-infection under iron-replete (blue) and -depleted (red) conditions. The transcript expression of *trpRBA* operon was quantified by RT-qPCR and normalized against genome copy number. Only *trpB* and *trpA* expression was significantly induced relative to 12 hpi. N = 3. Statistical significance was determined by One-Way ANOVA followed by post-hoc pairwise *t*-tests with Bonferroni's correction for multiple comparisons. *=$p < 0.05$, **=$p < 0.01$, ***=$p < 0.005$.

DOI: https://doi.org/10.7554/eLife.42295.008

The following source data is available for figure 3:

**Source data 1.** Source data for *Figure 3B–C*.
DOI: https://doi.org/10.7554/eLife.42295.009

starvation as the primary mechanism by which persistence develops in *C. trachomatis* following exposure to IFN-γ. However, these studies have routinely depended on prolonged treatment conditions that monitor the terminal effect of persistent development, as opposed to the immediate molecular events which may have important roles in the developmental fate of *Chlamydia*. As such, these studies may have missed the contribution of other IFN-γ-stimulated insults such as iron limitation.

To decouple Trp limitation from iron limitation and assess their relative contribution to regulating a critical pathway for responding to IFN-γ-mediated stress, we monitored the transcript expression of the *trpRBA* operon under brief Trp or iron starvation by RT-qPCR. Here again, we analyzed changes in transcript levels at the 18 hpi time-point and between the 12 hpi and 12 hpi + 6 hr time-points. This allowed us to determine if differences in expression could be accounted for by reduced, maintained or induced expression relative to 12 hpi. When starved for Trp for 6 hr, we observed that the expression of *trpR*, *trpB* and *trpA* were all significantly induced greater than 5.18-fold relative to 12 hpi (p=0.0040, 0.020 and 0.0036, respectively; *Figure 3B*). All three ORFs were also significantly elevated relative to the equivalent mock-treated time-point (p=0.0039, 0.019 and 0.0035, respectively). This result demonstrated that a relatively brief duration of Trp starvation was sufficient to induce *trpRBA* transcription and highlights the highly attuned sensitivity of *C. trachomatis* to even moderate changes in Trp levels.

We then performed the same RT-qPCR analysis on the expression of the *trpRBA* operon in response to 6 hr of iron limitation via BpdI treatment (*Figure 3C*). While we observed that the transcript expression of all three ORFs was significantly elevated at least 2.1-fold relative to the

equivalent mock-treated time-point (p=0.015, 0.00098 and 0.0062, respectively), we made the intriguing observation that only the expression of *trpB* and *trpA* was significantly induced relative to 12 hpi (p=0.00383 and 0.0195, respectively). The marginal elevation in *trpR* expression at the 18 hpi time-point was surprising given that this gene was not identified as iron-responsive in a recent genome-wide RNA-sequencing study (*Brinkworth et al., 2018*). Our results suggested that while the *trpRBA* operon is responsive to iron limitation, *trpBA* may have a more complex mode of regulation given the additional induction observed relative to *trpR*, which only maintained expression between 12 hpi and 12 hpi +6 hr Bpdl time-points. Taken together, these findings demonstrate that an important stress response pathway, the *trpRBA* operon, is regulated by the availability of both Trp and iron, consistent with the notion that the pathway may be cooperatively regulated to respond to various stress conditions. Notably, iron-dependent regulation of Trp biosynthesis has not been previously documented in other prokaryotes.

## Specific iron-regulated expression of *trpBA* originates from a novel alternative transcriptional start site within the *trpRBA* intergenic region

We hypothesized that the specific iron-related induction of *trpBA* expression relative to *trpR* expression may be attributable to an iron-regulated alternative transcriptional start site (alt. TSS) downstream of the *trpR* ORF. Indeed, a previous study reported the presence of an alt. TSS in the *trpRBA* IGR, located 214 nucleotides upstream of the *trpB* translation start position (*Carlson et al., 2006*). However, a parallel study could not identify a TrpR binding site in the *trpRBA* IGR (*Akers and Tan, 2006*). We reasoned that a similar alt. TSS may exist in the IGR that controlled the iron-dependent expression of *trpBA*. We therefore performed Rapid Amplification of 5′-cDNA Ends (5′-RACE) on RNA isolated from *C. trachomatis* L2-infected HeLa cells using the SMARTer 5′/3′ RACE Kit workflow (Takara Bio). Given the low expression of the *trpRBA* operon during normal development, we utilized two sequential gene-specific amplification steps (nested 5′-RACE) to identify 5′ cDNA ends in the *trpRBA* operon. These nested RACE conditions resulted in amplification that was specific to infected-cells (*Figure 4—figure supplement 1A*). Using this approach, we analyzed four conditions: 12 hpi, 18 hpi, 12 hpi + 6 hr of Bpdl treatment, and 12 hpi + 6 hr of Trp-depletion (*Figure 4A*). We observed three RACE products that migrated with an apparent size of 1.5, 1.1 and 1.0 kilobases (kb). At 12 and 18 hpi, all three RACE products exhibited low abundance, even following the nested PCR amplification. This observation was consistent with the expectation that the expression of the *trpRBA* operon is very low under normal, iron and Trp-replete conditions. However, we note that the 6 hr difference in development did appear to alter the representation of the 5′ cDNA ends, which may suggest a stage-specific promoter utilization within the *trpRBA* operon. In our Trp starvation condition, we observed an apparent increase in the abundance of the 1.5 kb RACE product, which was therefore presumed to represent the primary TSS upstream of *trpR*, at nucleotide position 511,389 (*C. trachomatis* L2 434/Bu). Interestingly, the 1.0 kb product displayed a very similar apparent enrichment following Bpdl treatment, suggesting that this RACE product represented a specifically iron-regulated TSS. Both the 1.5 and 1.0 kb RACE products were detectable in the Trp-depleted and iron-depleted conditions, respectively, during the primary RACE amplification, consistent with their induction under these conditions (*Figure 4—figure supplement 1B*).

If iron depletion was inducing *trpBA* expression independent of *trpR*, we reasoned that we would observe specific enrichment of *trpB* transcripts in our 5′-RACE cDNA samples relative to *trpR* transcripts. We again utilized RT-qPCR to quantify the abundance of *trpB* transcripts relative to *trpR* transcripts in the 5′-RACE total RNA samples (*Figure 4B*). In agreement with our model, only under iron starved conditions did we observe a significant enrichment of *trpB* relative to *trpR* (p<0.01). Additionally, we observed that at 12 and 18 hpi in iron-replete conditions, the ratio of *trpB* to *trpR* was approximately 1.0, suggesting non-preferential basal expression across the three putative TSSs. Another factor contributing to this ratio is the synthesis of the full-length *trpRBA* polycistron. In support of this, the *trpB* to *trpR* ratio remained near 1.0 under the Trp-starved condition, which would be expected during transcription read-through of the whole operon. The apparent lack of preferential promoter utilization as described above could be attributed to the relatively low basal expression of the operon at 12 and 18 hpi under Trp- and iron-replete conditions, thus precluding quantitative detection of differential promoter utilization in this assay.

To determine the specific location of the 5′ cDNA ends within the *trpRBA* operon, we isolated the 5′-RACE products across all conditions by gel extraction and cloned the products into the

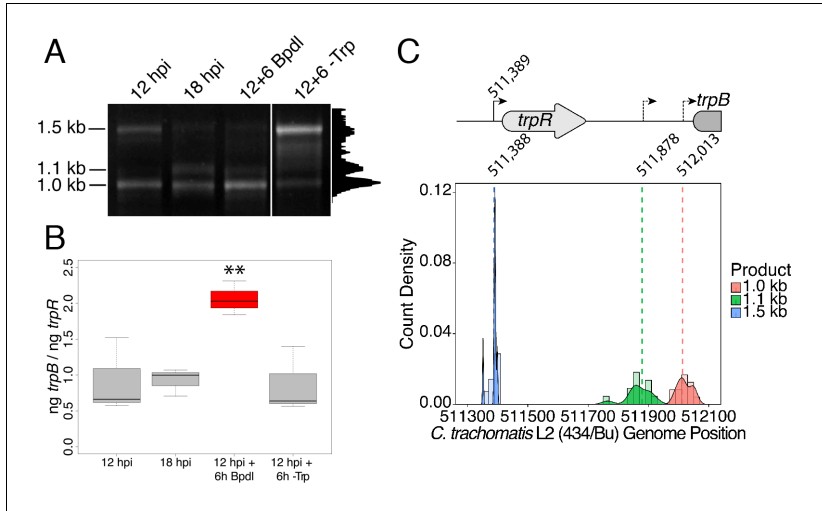

**Figure 4.** Iron-dependent induction of *trpBA* expression initiates within the *trpRBA* intergenic region from a novel alternative transcriptional start site. (**A**) Total RNA was harvested from infected HeLa cells at the indicated times post-infection to examine iron-dependent and Trp-dependent changes in the 5'-cDNA profile of the *trpRBA* operon by Rapid Amplification of 5' cDNA Ends (5'-RACE). RACE products were separated on an agarose gel, revealing three distinct and specific bands with apparent sizes of 1.5, 1.1 and 1.0 kb. Trp depletion led to the apparent enrichment of the 1.5 kb product, while BpdI treatment produced a similarly enriched 1.0 kb RACE product. Intensity plot to the right of image was generated using the Fiji Dynamic ROI Profiler plugin to monitor intensity across the 18 hpi condition. Figure shows representative experiment of three biological replicates. (**B**) To confirm that iron-dependent induction of *trpBA* could originate from alternative transcription initiation, RT-qPCR was performed on 5'-RACE total RNA to quantify the abundance of *trpB* transcripts relative to *trpR*. Only under iron-limited conditions were *trpB* transcripts enriched relative to *trpR*. N = 3. Statistical significance determined by One-way ANOVA followed by post-hoc pairwise *t*-tests. *=$p < 0.05$, **=$p < 0.01$, ***=$p < 0.005$. (**C**) The nucleotide position of the 5' cDNA ends generated from RACE were mapped to the *C. trachomatis* L2 434/Bu genome by nucleotide BLAST. Figure displays histogram (semi-continuous; bin width = 20) and overlaid density plot (continuous) distribution of 5' nucleotide positions generated from each 5'-RACE product. The dotted line represents the weighted mean of the distribution, as indicated by the integer value above each line. The identified alt. TSSs are depicted on the *trpRBA* operon (drawn to scale) above the plot. At least four clones were sequenced for each RACE product per replicate. N = 3.

DOI: https://doi.org/10.7554/eLife.42295.010

The following source data and figure supplements are available for figure 4:

**Source data 1.** Source data for *Figure 4B*.
DOI: https://doi.org/10.7554/eLife.42295.013
**Figure supplement 1.** 5'-RACE conditions produce *Chlamydia*-specific products that are amplified in primary RACE.
DOI: https://doi.org/10.7554/eLife.42295.011
**Figure supplement 2.** Mapping of 5'-RACE products at the individual nucleotide level.
DOI: https://doi.org/10.7554/eLife.42295.012

pRACE vector supplied by the manufacturer. We then sequenced the ligated inserts and BLASTed the sequences against the *C. trachomatis* L2 434/Bu genome to identify the location of the 5'-most nucleotides (*Figure 4C*). These data are displayed as a statistical approximation of the genomic regions most likely to be represented by the respective 5'-RACE products in both histogram (semi-continuous) and density plot (continuous) format (See *Supplementary file 1* for a description of all mapped 5'-RACE products). As expected, the 1.5 kb product mapped in a distinct and tightly grouped peak near the previously annotated *trpR* TSS, with the mean and modal nucleotide being 511,388 and 511,389, respectively (*Figure 4—figure supplement 2A*). Surprisingly, we found that neither the 1.1 or 1.0 kb RACE product mapped to the previously reported alt. TSS in the *trpRBA* IGR, at position 511,826. Instead, we observed that the 1.1 kb product mapped on average to nucleotide position 511,878, with the modal nucleotide being found at 511,898 (*Figure 4—figure supplement 2B*). The 1.0 kb product mapped with a mean nucleotide position of 512,013, with the modal

nucleotide being 512,005 (*Figure 4—figure supplement 2C*), only 35 bases upstream of the *trpB* coding sequence. Interestingly, the 1.0 kb product mapped to a region of the *trpRBA* IGR flanked by consensus $\sigma^{66}$ -10 and −35 promoter elements, found at positions 512,020–5 and 511,992–7, respectively (*Ricci et al., 1995*). In *C. trachomatis*, $\sigma^{66}$ is the major housekeeping sigma factor, homologous to *E. coli* $\sigma^{70}$. In silico analyses did not reveal the presence of any promoter elements near the 1.1 kb product, however the mean nucleotide position is 50 bp downstream of the previously identified palindrome suspected to have a TrpR operator function (*Carlson et al., 2006*). These data collectively pointed toward the 1.0 kb 5'-RACE product representing a novel, iron-regulated alt. TSS and *bona fide* $\sigma^{66}$-dependent promoter element that allows for the specific iron-dependent expression of *trpBA*.

## YtgR specifically binds to the *trpRBA* intergenic region in an operator-dependent manner to repress transcription of *trpBA*

As the only known iron-dependent transcriptional regulator in *Chlamydia*, we hypothesized that YtgR may regulate the iron-dependent expression of *trpBA* from the putative promoter element we characterized by 5'-RACE. Using bioinformatic sequence analysis, we investigated whether the *trpRBA* IGR contained a candidate YtgR operator sequence. By local sequence alignment of the putative YtgR operator sequence (*Akers et al., 2011*) and the *trpRBA* IGR, we identified a high-identity alignment (76.9% identity) covering 67% of the putative operator sequence (*Figure 5A*). Interestingly, this alignment mapped to the previously identified palindrome suspected to have operator functionality (*Carlson et al., 2006*). By global sequence alignment of the YtgR operator to the palindromic sequence, an alignment identical to the local alignment was observed, which still displayed relatively high sequence identity (43.5% identity). We hypothesized that this sequence functioned as a YtgR operator, despite being located 184 bp upstream of the *trpBA* alt. TSS.

To investigate the ability of YtgR to bind and repress transcription from the putative *trpBA* promoter, we implemented a heterologous in vivo two-plasmid assay that reports on YtgR repressor activity as a function of β-galactosidase expression (*Thompson et al., 2012*). In brief, a candidate DNA promoter element was cloned into the pCCT expression vector between an arabinose-inducible pBAD promoter and the reporter gene *lacZ*. This plasmid was co-transformed into BL21 (DE3) *E. coli* along with an IPTG-inducible pET151 expression vector with (pET151-YtgR) or without (pET151-EV) the C-terminal 139 amino acid residues of CTL0325 (YtgC). Note that we have previously demonstrated that this region is a functional iron-dependent repressor domain (*Thompson et al., 2012*). To verify the functionality of this assay, we determined whether ectopic YtgR expression could repress pCCT reporter gene expression in the presence of three candidate DNA elements: a no-insert empty vector (pCCT-EV), the putative promoter element for *C. trachomatis dnaB* (pCCT-*dnaB*), and the promoter region of the *ytgABCD* operon (pCCT-*ytgABCD*; *Figure 5B*). As expected, from the pCCT-EV reporter construct, ectopic YtgR expression did not significantly reduce the activity of β-galactosidase. Additionally, reporter gene expression from pCCT-*dnaB*, containing the promoter of non-iron-regulated *dnaB*, was not affected by ectopic expression of YtgR. In contrast, in the presence of pCCT-*ytgABCD*, induction of YtgR expression produced a significant decrease in β-galactosidase activity (p=0.03868) consistent with its previously reported auto-regulation of this promoter (*Thompson et al., 2012*).

Using this same assay, we then inserted into the pCCT reporter plasmid 1) the *trpR* promoter element (pCCT-*trpR*), 2) the putative *trpBA* promoter element represented by the IGR (pCCT-*trpBA*), and 3) the same putative *trpBA* promoter element with a mutated YtgR operator sequence that was diminished for both palindromicity and A-T richness, two typical features of prokaryotic promoter elements (pCCT-*trpBA*ΔOp; *Figure 5C*) (*Schmitt, 2002*; *Tao et al., 1992*). Note that the *trpRBA* IGR is >99% conserved at the nucleotide level across urogenital, ocular and LGV serovars of *C. trachomatis*, and the putative YtgR operator sequence is 100% identical (*Figure 5—figure supplement 1*). When YtgR was ectopically expressed in the pCCT-*trpR* background, we observed no statistically distinguishable change in β-galactosidase activity, indicating YtgR could not regulate transcription from the *trpR* promoter. However, in the pCCT-*trpBA* background, ectopic YtgR expression significantly reduced β-galactosidase activity at levels similar to those observed in the pCCT-*ytgABCD* background (p=0.01219). This suggested that YtgR was capable of repressing transcription from the *trpBA* promoter element specifically. Interestingly, this repression phenotype was abrogated in the pCCT-*trpBA*ΔOp background, where we observed no statistically meaningful difference in β-

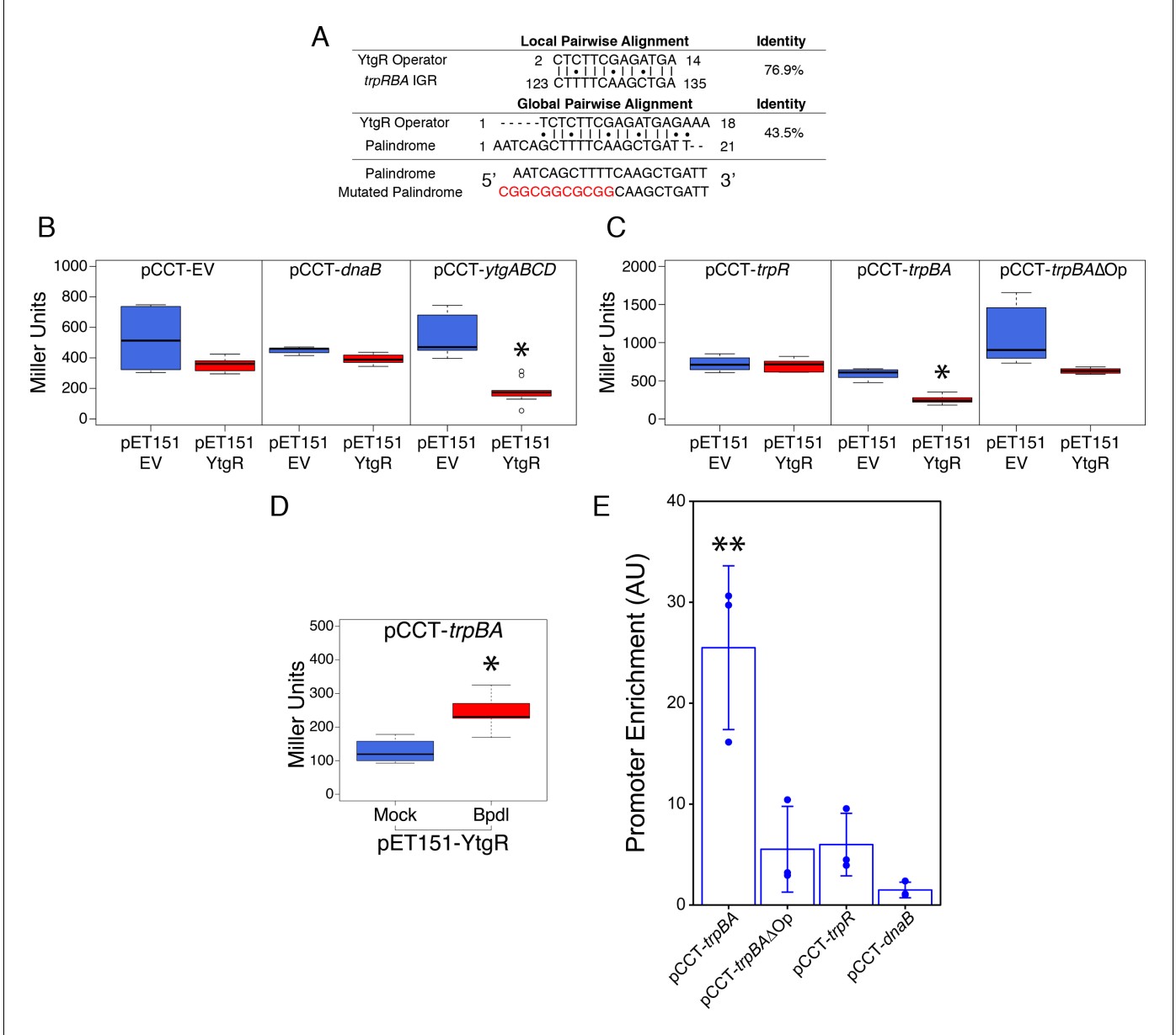

**Figure 5.** Ectopically expressed YtgR is binds the putative *trpBA* promoter element in an operator-specific manner to repress transcription in a heterologous in vivo system. (**A**) Identification of putative YtgR operator sequence by local and global nucleotide sequence alignment using EMBOSS Water and Needle algorithms, respectively, to align the previously identified YtgR operator to both the *trpRBA* IGR and palindromic candidate sequence. The palindrome was then mutated in our YtgR repression assay as depicted to abolish palindromicity and AT-richness. (**B**) Ectopic expression of YtgR significantly represses β-galactosidase activity only from the promoter of its own operon, *ytgABCD*, and not from an empty vector or the non-iron-regulated promoter of *dnaB*. N = 3. (**C**) Expression of recombinant YtgR represses β-galactosidase activity from the putative *trpBA* promoter element, but not the *trpR* promoter, and this repression is dependent on the unaltered operator sequence identified in *Figure 5A*. (**D**) Bipyridyl-treatment reduces YtgR repressor activity at the *trpBA* promoter element, consistent with the iron-dependent DNA-binding of YtgR. (**E**) Immunoprecipitation of YtgR reveals specific and direct interaction with the *trpBA* promoter that is dependent upon the native operator sequence. N = 3. For the Miller assay, statistical significance was determined by two-sided unpaired Student's *t*-test with Welch's correction for unequal variance. For the ChIP assay, statistical significance was determined by One-Way ANOVA and post-hoc pairwise *t*-test with Bonferroni's correction for multiple comparisons. *=$p < 0.05$, **=$p < 0.01$, ***=$p < 0.005$.

DOI: https://doi.org/10.7554/eLife.42295.014

The following source data and figure supplements are available for figure 5:

**Source data 1.** Source data for *Figure 5B–E*.

DOI: https://doi.org/10.7554/eLife.42295.017

*Figure 5 continued on next page*

*Figure 5 continued*

**Figure supplement 1.** The *trpRBA* IGR is greater than 99% conserved across ocular, genital and LGV *C.*
DOI: https://doi.org/10.7554/eLife.42295.015
**Figure supplement 2.** Truncated fragments of the *trpRBA* IGR are not sufficient to confer YtgR repression phenotype, regardless of the presence of the putative operator site.
DOI: https://doi.org/10.7554/eLife.42295.016

galactosidase activity, demonstrating that YtgR transcriptional repression of *trpBA* is operator-dependent. We subsequently addressed whether the region of the *trpRBA* IGR containing the YtgR operator site was sufficient to confer YtgR repression in this assay (*Figure 5—figure supplement 2*). We cloned three fragments of the *trpRBA* IGR into the pCCT reporter plasmid: the first fragment represented the 5'-end of the IGR containing the operator site at the 3'-end (pCCT-IGR1), the second fragment represented a central region of the IGR containing the operator site at the 5'-end (pCCT-IGR2), and the third fragment represented the 3'-end of the IGR and did not contain the operator site (pCCT-IGR3). Surprisingly, we observed that none of these fragments alone were capable of producing a significant repression phenotype in our reporter system.

To verify that YtgR repression of the *trpBA* promoter element was dependent upon iron availability, we assessed repressor activity in the presence or absence of 500 µM BpdI, mimicking previously utilized approaches for assessing iron-dependent repressor activity of the homologous DtxR (*Ding et al., 1996*). YtgR is known to bind cognate DNA elements in an iron-dependent fashion (*Thompson et al., 2012*), and as such any repressor activity of YtgR should be a direct consequence of its ability to bind DNA in the presence of iron. When we treated co-transformed *E. coli* expressing YtgR and harboring the pCCT-*trpBA* reporter plasmid, we observed a modest but statistically significant increase in β-galactosidase activity (*Figure 5D*; p=0.01409), consistent with the alleviation of YtgR repression at the *trpBA* promoter element.

To demonstrate that the repression phenotype observed in this reporter system was attributable to DNA-binding of YtgR, we optimized a targeted chromatin immunoprecipitation (ChIP) qPCR method to detect the abundance of co-immunoprecipitated promoter fragments with the recombinant YtgR domain. As before, we co-transformed the pET151-YtgR expression vector with pCCT-*trpBA*, pCCT-*trpBA*ΔOp, pCCT-*trpR* or pCCT-*dnaB* and then fixed the co-transformed cells with formaldehyde prior to immunoprecipitation of the cross-linked 6xHis-YtgR-DNA complexes. Using this system, we observed a significant enrichment of *trpBA* promoter relative to the *dnaB* negative control promoter (p=0.0018; *Figure 5E*). However, enrichment of the *trpR* promoter was marginal and not statistically distinguishable from that of *dnaB*. Consistent with its requirement for repression, enrichment of *trpBA*ΔOp was also marginal and statistically indistinguishable from *trpR* or *dnaB*, suggesting that mutation of the putative operator sequence alone is sufficient to abrogate YtgR DNA-binding to the *trpBA* promoter. In conjunction with the IGR fragment analysis, these findings indicated that while the operator site was necessary for YtgR DNA-binding and transcriptional repression, further unknown structural elements in the *trpRBA* IGR may be required for repression. Nonetheless, this demonstrated the existence of a functional YtgR binding site that conferred transcriptional regulation to *trpBA*, independent of the major *trpR* promoter. Collectively, the most parsimonious model derived from these data is that specific and direct iron-dependent DNA-binding of YtgR at the identified operator site acts to repress expression of genes downstream of the *trpBA* promoter element.

## Transcripts initiated at the primary *trpR* promoter terminate at the YtgR operator site

We hypothesized that YtgR binding at the *trpRBA* YtgR operator site may disadvantage the processivity of RNAP reading-through the IGR from the upstream *trpR* promoter, possibly leading to transcript termination. Similar systems of RNAP read-through blockage have been reported; the transcription factor Reb1p 'roadblocks' RNAPII transcription read-through in yeast by promoting RNAP pausing and subsequent degradation (*Colin et al., 2014*). To investigate this question, we first returned to RNA-Sequencing data we generated to define the immediate iron-dependent transcriptional regulon in *C. trachomatis* (*Brinkworth et al., 2018*). Using data obtained from *C. trachomatis*-infected HeLa cells at 12 hpi +6 hr mock or BpdI treatment, we mapped the sequenced reads in

batch across three biological replicates to the *C. trachomatis* L2 434/Bu genome (NC_010287) which we modified to include annotations for non-operonic IGRs. Using this coverage map, we sought to semi-quantitatively assess the mapping of reads across the *trpRBA* IGR to gain insight regarding possible transcription readthrough. Our analysis revealed that under Bpdl-treated conditions, there

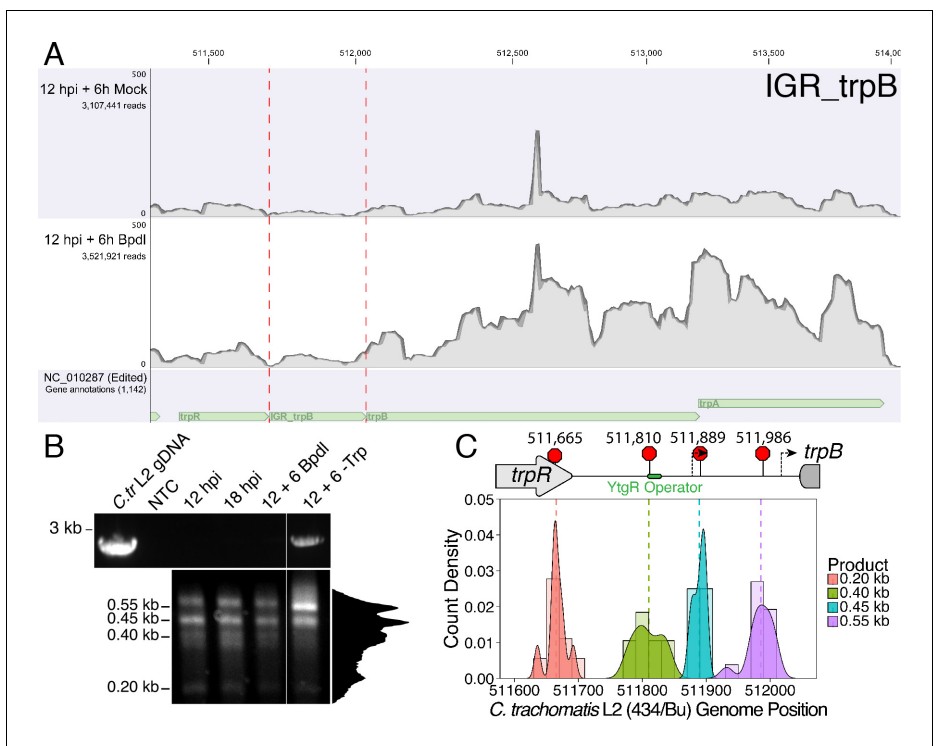

**Figure 6.** Transcription from the primary *trpR* promoter terminates in the *trpRBA* IGR at the YtgR operator site. (**A**) Coverage map of RNA-Sequencing reads mapped to the *C. trachomatis* L2/434 Bu genome (NC_010287) edited to contain annotations for IGRs. Read coverage at the *trpRBA* IGR (IGR_trpB) is increased following Bpdl treatment, but *trpR* read coverage is not similarly increased. (**B**) Total RNA was harvested from *C. trachomatis*-infected HeLa cells to analyze transcript termination landscape downstream of the *trpR* promoter by 3'-RACE. The top panel displays representative RT-PCR of full-length *trpRBA* message across experimental conditions (NTC = No Template Control). Bottom panel depicts electrophoresed 3'-RACE products and estimated sizes. Intensity plot to the right of image was generated using the Fiji Dynamic ROI Profiler plugin to monitor intensity across the 18 hpi condition. Note the presence of four distinct peaks, corresponding to each 3'-RACE product. N = 3. (**C**) 3'-RACE products were sequenced and mapped to the *C. trachomatis* L2 434/Bu genome by nucleotide BLAST. The figure displays histogram (semi-continuous; bin width = 20) and overlaid density plot (continuous) distribution of 3' nucleotide positions generated from each 3'-RACE product. The dotted line represents the weighted mean of the distribution, as indicated by the integer value above each line. The identified alt. TTSs are depicted on the *trpRBA* operon (drawn to scale) above the plot. At least four clones were sequenced for each RACE product per replicate. N = 3.

DOI: https://doi.org/10.7554/eLife.42295.018

The following figure supplements are available for figure 6:

**Figure supplement 1.** RNA-Sequencing coverage map of reads mapping to the intergenic regions upstream of *euo* and *lpdA*.

DOI: https://doi.org/10.7554/eLife.42295.019

**Figure supplement 2.** RPKM Fold change analysis from iron-starved RNA-Sequencing data.

DOI: https://doi.org/10.7554/eLife.42295.020

**Figure supplement 3.** 3'-RACE conditions produce *Chlamydia*-specific products that are amplified in primary RACE.

DOI: https://doi.org/10.7554/eLife.42295.021

**Figure supplement 4.** Mapping of the 3'-RACE products at the individual nucleotide level.

DOI: https://doi.org/10.7554/eLife.42295.022

was a 2.21-fold increase in reads mapping to the *trpRBA* IGR (IGR_trpB) relative to mock treatment (*Figure 6A*). This observation is consistent with the alleviation of YtgR repression under iron-starved conditions permitting readthrough of transcription from the upstream *trpR* promoter. However, due to high variation and the low number of reads mapping to this region, we were unable to detect a significant difference in coverage (p=0.11) using the genomewide RNA-seq analysis toolkit in the CLC Genomics Workbench. Regardless, a comparable increase in reads mapping to the upstream *trpR* CDS was not observed (1.54-fold increase, p=0.32), suggesting that under iron replete conditions, transcripts originating from the primary *trpR* promoter may be terminated before reading through the IGR, thereby accounting for the increase in reads mapping to IGR_trpB following BpdI treatment.

We additionally assessed the read coverage of the IGRs upstream of *euo* (IGR_euo; not iron-regulated) and *lpdA* (IGR_lpdA; iron-regulated, *Brinkworth et al., 2018*) which are similarly configured compared to the *trpRBA* IGR (*i.e.* the IGRs are between two ORFs in the same coding orientation). For both IGR_euo (1.17-fold, p=0.58) and IGR_lpdA (1.40-fold, p=0.37), we did not observe a similar increase in read coverage following BpdI treatment indicating that the increased coverage at IGR_trpB is likely specific (*Figure 6—figure supplement 1A–B*). Qualitatively, we note that only IGR_trpB displays relatively uniform read coverage across the defined region, while both IGR_euo and IGR_lpdA have non-uniform coverage, reinforcing the idea that the reads mapping to IGR_trpB originate upstream in the *trpR* ORF and readthrough the entire region, thus presenting the opportunity for premature termination. Furthermore, the upstream ORFs for IGR_euo and IGR_lpdA did not display robust fold-increases; *recJ*, upstream of IGR_euo, was 1.29-fold increased (p=0.17) while *CTL0819*, upstream of IGR_lpdA, was 1.13-fold increased (p=0.56). We extracted individual RPKM values for each of these regions from this dataset and observed the same trend in mean fold-change differences reported from the genomewide analysis: only IGR_trpB was substantially increased relative to its upstream ORF (*Figure 6—figure supplement 2*). Despite being informative, genomewide RNA-Seq is ultimately insufficient to elucidate particular mechanistic details of transcriptional regulation in the *trpRBA* IGR and therefore we turned to more sensitive and quantitative methods to investigate possible transcript termination within the *trpRBA* IGR.

To identify transcription termination sites (TTSs) in the *trpRBA* operon in *C. trachomatis*, we utilized 3'-RACE to map the 3'-ends of transcripts using gene-specific primers within the *trpR* CDS (*Figure 6*; lower panel). We again utilized two RACE amplification cycles to generate distinct, specific bands suitable for isolation and sequencing (*Figure 6—figure supplement 3B–C*). By gel electrophoresis of the 3'-RACE products, we observed the appearance of four distinct bands that migrated with an apparent size of 0.55, 0.45, 0.40 and 0.20 kb. In our Trp-depleted condition, we observed only a very weak amplification of the 2.5–3 kb full-length *trpRBA* message by 3'-RACE (*Figure 6—figure supplement 3A*). However, we did observe it across all replicates. To confirm that the full-length product was specific to the Trp-depleted treatment, we amplified the *trpRBA* operon by RT-PCR from the 3'-RACE total RNA (*Figure 6B*; upper panel). As expected, only in the Trp-depleted sample did we observe robust amplification of the full-length *trpRBA* message. We note however that image contrast adjustment reveals a very weak band present in all experimental samples. Therefore, the specific 3'-RACE analysis identified novel transcription termination events within the *trpRBA* operon.

To identify the specific TTS locations, we gel extracted the four distinct 3'-RACE bands across all conditions and cloned them into the pRACE sequencing vector as was done for the 5'-RACE experiments. We then sequenced the inserted RACE products and mapped them to the *C. trachomatis* L2 434/Bu genome (*Figure 6C*). This revealed a highly dynamic TTS landscape within the *trpRBA* IGR, which has not previously been investigated (For a full description of mapped 3'-RACE products, see *Supplementary file 2*). The 0.20 kb RACE product mapped to the 3'-end of the *trpR* CDS, with a mean nucleotide position of 511,665 and a modal nucleotide position of 511,667 (*Figure 6—figure supplement 4A*). Contrastingly, the other three 3'-RACE products did not map in such a way so as to produce specific, unambiguous modal peaks. Instead, their distribution was broader and more even, with only a few nucleotide positions mapping more than once. Accordingly, the 0.45 kb product mapped with an average nucleotide position of 511,889, just downstream of the 1.1 kb 5'-RACE product (*Figure 6—figure supplement 4C*), while the 0.55 kb product mapped with an average nucleotide position of 511,986, upstream of the 1.0 kb 5'-RACE product (*Figure 6—figure supplement 4D*). Interestingly, the 0.40 kb product mapped to a region directly overlapping the putative

YtgR operator site, with a mean nucleotide position of 511,810 (*Figure 6—figure supplement 4B*). We therefore reasoned that this putative TTS may have an iron-dependent function controlled by YtgR.

## YtgR mediates iron-dependent termination of upstream transcripts at the putative *trpRBA* operator site

We hypothesized that transcript termination at the YtgR operator site was regulated in an iron-dependent manner by YtgR binding to the operator DNA and blocking upstream transcription read-through. Under iron-replete conditions, YtgR would be bound to the operator DNA and transcript termination would occur at a greater frequency, preventing readthrough of the transcription machinery initiated at the upstream *trpR* promoter. Under iron-depleted conditions, the inactivation of YtgR DNA-binding activity would allow the transcription machinery to readthrough the YtgR operator site to the downstream sequence, including *trpBA*. To test this model, we utilized RT-qPCR to quantify the amount of readthrough at the YtgR operator site in iron-depleted *C. trachomatis*-infected HeLa cells and in our two-plasmid *E. coli* system where the expression of YtgR could be controlled.

We tested whether nutrient availability altered the extent of readthrough in the IGR at the YtgR operator site using a RT-qPCR-based quantification of various mRNA species (*Figure 7A*). Levels of each intermediate transcript species indicated by unique non-overlapping amplicons were reported as ratios against a common upstream amplicon. Thus, for amplicons downstream of termination sites, a higher downstream amplicon-to-common amplicon ratio would indicate increased read-through. Note that this value has a theoretical limit of 1.0, where all transcripts would be at least as long as the downstream amplicon being measured. We designed primers to amplify regions immediately 5' and 3' to the YtgR operator and its corresponding TTS. Additionally, we analyzed the very 3' end of *trpA*, which would be expected to only monitor complete full-length transcripts as well as alternative transcription initiation from the *trpBA* promoter.

First, we utilized our infected cell-culture model to assess the iron-dependency of transcription readthrough at the YtgR operator site. As before, we assayed four conditions: 12 hpi, 18 hpi, 12 hpi +6 hr Bpdl and 12 hpi +6 hr Trp-depletion. Under normal developmental conditions, we expected that there would not be dramatic changes in readthrough, though we note that changes in development or nutrient availability could have this affect under normal conditions. Under Trp-depleted conditions, we predicted that the inactivation of the TrpR repressor would produce a robust readthrough phenotype that, in accordance with the literature and evidence provided herein, would transcribe the full-length *trpRBA* message. Thus, Trp-depletion served as a positive control for our readthrough analysis. We hypothesized that under iron-depleted conditions, we would not observe a significant increase in readthrough 5' of the YtgR operator site, but when YtgR is inactivated, we should be able to detect an increase in readthrough 3' of the operator site as the transcription machinery is no longer blocked by the bound repressor and is permitted to continue transcription into the downstream sequence.

Consistent with our model, readthrough 5' of the YtgR operator site was not affected by Bpdl treatment, whereas Trp-depletion resulted in significantly more readthrough than all other conditions (*Figure 7B*; p<0.05 for all comparisons). However, 3' of the YtgR operator site, Bpdl treatment significantly increased readthrough compared to 12 and 18 hpi (p<0.05 for all comparisons), suggesting that inactivation of YtgR by iron depletion alleviates transcription termination at this site. We performed this same analysis with an alternative amplicon 3' of the YtgR operator site and observed a similar increase in readthrough, confirming that the effect was not unique to the amplicon we used (*Figure 7—figure supplement 1*; p<0.005 for all comparisons). When this readthrough analysis is applied 3' of the alternative *trpB* TSS, we observe that following Bpdl-treatment, the readthrough value substantially exceeds 1.0, consistent with transcription of *trpBA* independent of *trpR* under this condition. Moreover, only the Bpdl-treated group is significantly different from the other treatment conditions (p<0.05 for all comparisons). Notably, Trp-depletion did not increase the read-through value, suggesting that Trp-depletion does not relieve YtgR repression at the alt. TSS and the levels of *trpA* remain relatively constant to those of *trpR*. It is therefore possible that under Trp-depleted, but iron-replete conditions, YtgR repression of the alternative *trpBA* promoter acts as a rheostat for full-length *trpRBA* transcription, helping to maintain a constant ratio of *trpR* to *trpBA*.

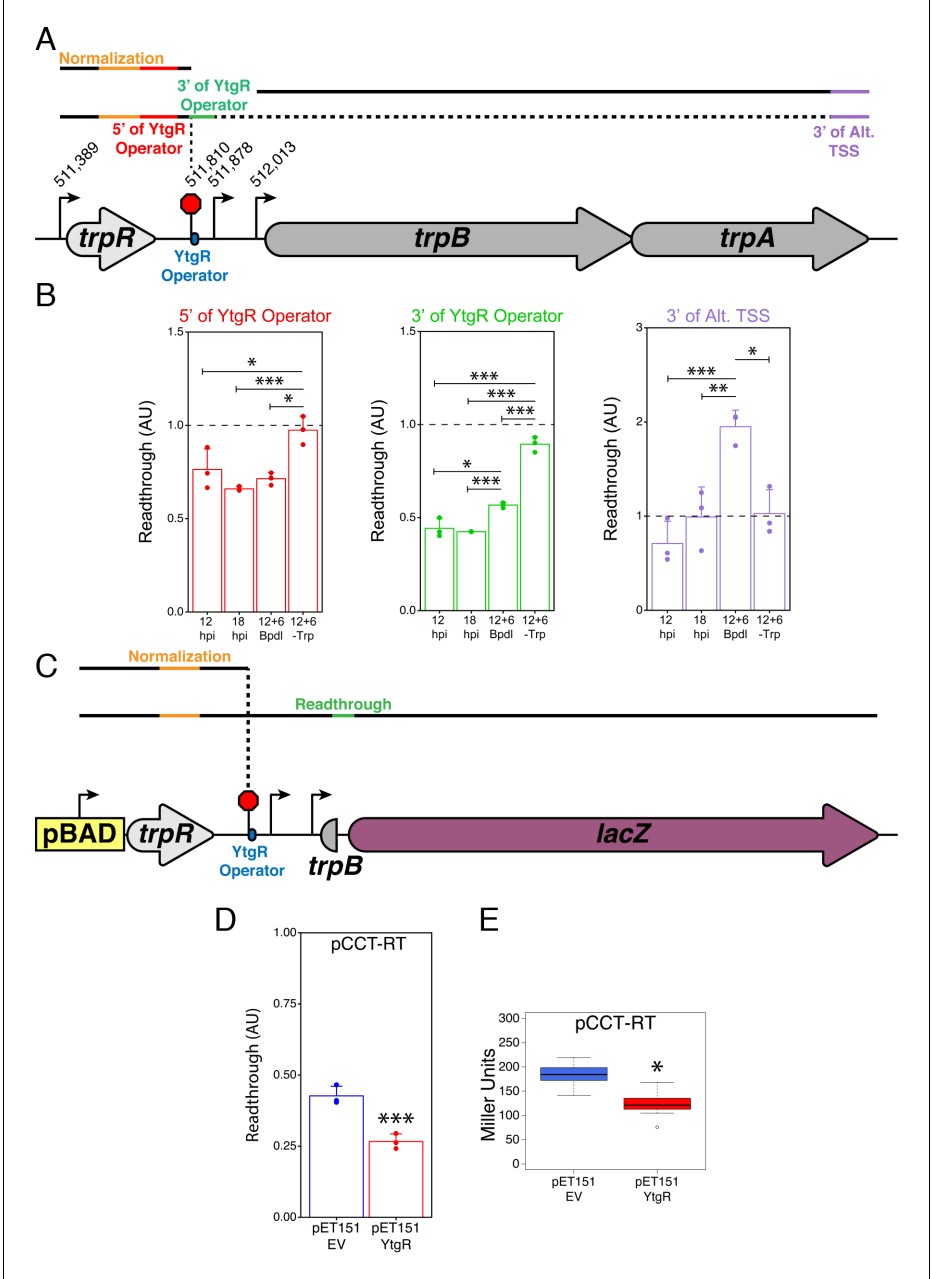

**Figure 7.** YtgR mediates iron-dependent transcriptional blockage at the putative operator site in the *trpRBA* IGR. (**A**) Graphical schematic of RT-qPCR amplicons utilized to assess transcriptional readthrough in *C. trachomatis*-infected HeLa cells at the YtgR operator site. Black lines represent possible transcript species. Colored segments of line indicate unique amplicons. The dotted line indicates that the transcripts reading through the YtgR operator site may prematurely terminate at another site or readthrough the entire operon. (**B**) RT-qPCR analysis of transcription readthrough at the YtgR operator site in *C. trachomatis*-infected HeLa cells. Following Bpdl-treatment, readthrough is increased relative to mock-treated cultures 3' of the YtgR operator site. (**C**) Graphical schematic of pCCT-RT vector and RT-qPCR amplicons utilized to assess transcriptional readthrough in co-transformed BL21(DE3) *E. coli*. Black lines represent possible transcript species. Colored segments of line indicate unique amplicons. Note that the *lacZ* ORF is not drawn to scale. (**D**) RT-qPCR analysis of transcription readthrough of the pCCT-RT insert in the presence (pET151-YtgR) or absence (pET151-EV) of ectopically expressed recombinant YtgR demonstrates that expression of YtgR significantly reduces transcriptional readthrough of the *trpRBA* IGR. (**E**) Ectopic expression of YtgR significantly represses β-galactosidase activity from the pCCT-RT vector as determined by the Miller Assay in co-transformed BL21(DE3) *E. coli*. For all experiments, N = 3. For single pairwise comparisons, statistical significance was determined by two-sided unpaired Student's *t*-test with

*Figure 7 continued on next page*

*Figure 7 continued*

Welch's correction for unequal variance. For multiple pairwise comparisons, statistical significance was determined by One-way ANOVA followed by post-hoc pairwise *t*-tests with Bonferroni's correction for multiple comparisons. *=$p < 0.05$, **=$p < 0.01$, ***=$p < 0.005$.

DOI: https://doi.org/10.7554/eLife.42295.023

The following source data and figure supplement are available for figure 7:

**Source data 1.** Source data for *Figure 7B and D–E*, *Figure 7—figure supplement 1*.

DOI: https://doi.org/10.7554/eLife.42295.025

**Figure supplement 1.** Alternative amplicon 3' of the YtgR operator site indicates confirmatory increase in transcription readthrough.

DOI: https://doi.org/10.7554/eLife.42295.024

Together, these analyses indicated that iron limitation resulted in transcription readthrough specifically at the YtgR operator site.

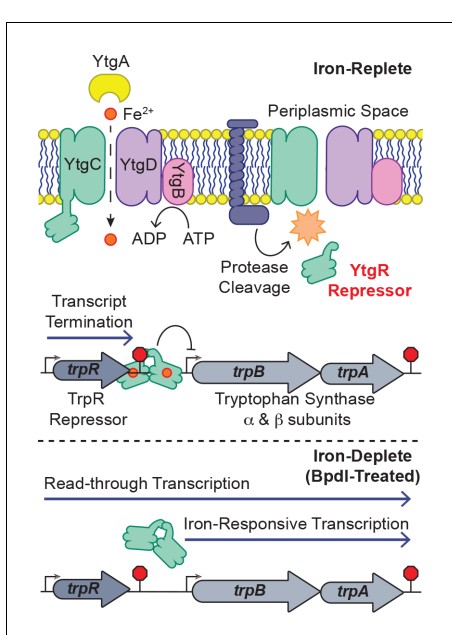

**Figure 8.** Model for proposed mechanism of iron-dependent YtgR-mediated regulation of *trpRBA* expression. Iron is imported through the YtgABCD ABC-type metal permease complex. YtgR is cleaved from the YtgCR permease-repressor fusion protein. In the presence of sufficient iron, holo-YtgR can bind to the *trpRBA* IGR to both terminate basal transcription from the primary *trpR* promoter and repress transcription initiation at the alternative *trpBA* promoter. Iron depletion inactivates YtgR DNA-binding, thus promoting read-through of basal transcription from the *trpR* promoter while also inducing transcription at the downstream *trpBA* promoter.

DOI: https://doi.org/10.7554/eLife.42295.026

The following figure supplement is available for figure 8:

**Figure supplement 1.** Comprehensive graphic of *trpRBA* T(S/T)S landscape.

DOI: https://doi.org/10.7554/eLife.42295.027

To investigate whether or not this readthrough phenotype was dependent upon YtgR, we again turned to our heterologous two-plasmid reporter system. We designed a reporter vector (pCCT-RT) that harbored the entire *trpR*-IGR DNA sequence such that any transcription initiated at the upstream arabinose-inducible pBAD promoter would have to readthrough the entire *trpR* ORF and the IGR before reaching the reporter gene *lacZ* (*Figure 7C*). Note that this is functionally similar to expression of the *trpRBA* operon under Trp-starved conditions: the major Trp-dependent promoter upstream of *trpR* would be activated and initiate readthrough independent of the presence of YtgR bound downstream in the IGR. As such, we performed the same RT-qPCR readthrough analysis on RNA harvested from BL21 (DE3) *E. coli* co-transformed with pCCT-RT and either pET151-EV or pET151-YtgR. We observed that ectopic expression of YtgR significantly reduced readthrough into the *lacZ* gene by RT-qPCR (*Figure 7D*; p=0.003561), consistent with YtgR DNA-binding specifically inhibiting readthrough via a mechanism of transcript termination at the operator site. In support of this result, we additionally observed a significant decrease in β-galactosidase activity as measured by the Miller assay from the pCCT-RT vector when YtgR was ectopically expressed (*Figure 7E*; p=0.01723), indicating that YtgR-dependent inhibition of readthrough limits the expression of downstream genes. In sum, YtgR binding to the *trpRBA* IGR at the predicted operator site concomitantly represses transcription from the alternative *trpBA* promoter while blockading transcription readthrough from the upstream *trpR* promoter, ultimately rendering the expression of *trpBA* susceptible to changes in iron availability. Importantly, alleviated transcript termination at the YtgR operator site may offer a

mechanistic explanation for the moderate elevation in *trpR* expression observed following iron star-vation, as enhanced readthrough at the YtgR operator site under this condition may produce more stable mRNA species relative to normally developing *C. trachomatis*. Thus, YtgR may function as an iron-dependent attenuator of *trpRBA* expression.

## Discussion

In this study, we provide a mechanistic explanation for the specific iron-limited induction of *trpBA* expression mediated by the repressor YtgR, representing a novel instance of integrated stress adap-tation in *Chlamydia*. Utilizing an infected-epithelial cell culture model, we identified a previously undescribed iron-regulated promoter element independent of *trpR* within the *trpRBA* IGR that is responsible for the iron-limited induction of *trpBA* expression. Using in silico and biochemical meth-ods, we demonstrate that YtgR binds the *trpRBA* IGR to regulate iron-dependent *trpBA* expression. Importantly, transcriptional repression in our heterologous system was shown to be dependent on an unaltered operator sequence that bears significant homology to the previously defined operator element in the *ytgA* promoter. Furthermore, our infected-cell culture studies revealed that tran-scripts originating from the primary *trpR* promoter terminate within the IGR, notably at the putative YtgR operator site, and that transcription read-through at this locus is iron- and YtgR-dependent. Thus, we propose that YtgR regulates *trpBA* expression at two levels: repression of the *trpBA* pro-moter and premature termination of the major transcript generated from the *trpR* promoter (**Fig-ure 8**; a comprehensive graphic of all T(S/T)Ss is provided in **Figure 8—figure supplement 1**). To our knowledge, this is the first time an iron-dependent mode of regulation has been shown to con-trol the expression of tryptophan biosynthesis in prokaryotes, which reflects the unique nature of *C. trachomatis*.

While we demonstrate here that iron-dependent *trpBA* expression originates from a novel pro-moter element immediately upstream of the *trpB* CDS, this is not the first description of an alt. TSS within the *trpRBA* IGR. **Carlson et al. (2006)** identified an alt. TSS within the IGR which they sug-gested was responsible for *trpBA* expression. In these studies, we were unable to confirm the pres-ence of the previously identified alt. TSS by 5'-RACE. This is likely because Carlson, *et al.* examined the presence of transcript origins following 24 hr of Trp starvation, whereas here we monitored immediate responses to stress following only 6 hr of treatment. Prolonged Trp depletion would result in a more homogeneously stressed population of chlamydial organisms that may exhibit the same preferential utilization of the promoter identified by Carlson, *et al.*, the detection of which is precluded in a more heterogeneous, transiently-stressed population. This may explain the observa-tion of multiple T(S/T)Ss across the *trpRBA* operon in our studies. However, the contribution of such a Trp-dependent alt. TSS as identified by Carlson *et al.* to the general stress response of *C. tracho-matis* remains unclear given its association with presumably abnormal organisms. Does utilization of this alt. TSS indicate abnormal growth or a *bona fide* stress adaptation? Moreover, Akers and Tan were unable to verify TrpR binding to the *trpRBA* IGR by EMSA, suggesting that some other Trp-dependent mechanism may control transcription from this site (**Akers and Tan, 2006**). Ultimately, our approach of investigating more immediate responses to stress revealed previously unreported mechanisms functioning to regulate Trp biosynthesis in *C. trachomatis*, underscoring the value of transient as opposed to sustained induction of stress.

Another mechanism of regulation reported to control the chlamydial *trpRBA* operon is Trp-dependent transcription attenuation. Based on sequence analysis, a leader peptide has been anno-tated within the *trpRBA* IGR (**Merino and Yanofsky, 2005**). Presumably, this functions analogously to the attenuator in the *E. coli trpEDCBA* operon; Trp starvation causes ribosome stalling at sites of enriched Trp codons such that specific RNA secondary structures form to facilitate RNAP read-through of downstream sequences – in the case of *C. trachomatis*, *trpBA* (**Yanofsky, 1981**). How-ever, robust experimental evidence to support the existence of attenuation in *C. trachomatis* is lacking. To date, the only experimental evidence that supports this model was reported by **Carlson et al. (2006)**, who demonstrated that in a TrpR-mutant genetic background, an additional increase in *trpBA* expression could be observed following 24 hr Trp-depletion. However, this could be attributable to an alternative Trp-dependent, but TrpR-independent mechanism controlling *trpBA* expression at the alt. TSS identified by Carlson, *et al*. None of the data presented here point conclusively to the existence of a Trp-dependent attenuator. The additional termination sites

identified in our 3'-RACE assay may represent termination events mediated by a Trp-dependent attenuator, but without more specific analysis utilizing mutated sequences we cannot attribute attenuator function to those termination sites. We cannot exclude the possibility that the other TTSs observed in our 3'-RACE analysis are iron-dependent, or the product of other forms of post-transcriptional regulation such as RNA processing, stability, etc. Our focus herein was to determine the YtgR-mediated mechanism of regulation, and we have defined at least one iron-dependent, YtgR-mediated termination site. Iron-dependent termination at other TTSs in the IGR would invariably produce the same effect of limiting *trpBA* expression under iron-replete conditions.

In *Bacillus subtilis,* Trp-dependent attenuation of transcription takes on a form markedly different from that in *E. coli*. Whereas attenuation functions in cis for the *E. coli trp* operon, *B. subtilis* utilize a multimeric Tryptophan-activated RNA-binding Attenuation Protein, TRAP, which functions in trans to bind *trp* operon RNA under Trp-replete conditions, promoting transcription termination and inhibiting translation (*Gollnick et al., 2005*). This interaction is antagonized by anti-TRAP in the absence of charged tRNA$^{Trp}$, leading to increased expression of TRAP regulated genes. We suggest that YtgR may represent the first instance of a separate and distinct clade of attenuation mechanisms: iron-dependent *trans*-attenuation. This mechanism may function independently of specific RNA secondary structure, relying instead on steric blockage of RNAP processivity, but ultimately producing a similar result. Possible regulation of translation remains to be explored. The recent development of new genetic tools to alter chromosomal sequences and conditionally knockdown gene expression in *C. trachomatis* should enable a more detailed analysis of *trpRBA* regulation, including possible *trans*-attenuation (*Keb et al., 2018*; *Mueller et al., 2016*; *Ouellette, 2018*).

As a Trp auxotroph, what might be the biological significance of iron-dependent YtgR regulation of the *trpRBA* operon in *C. trachomatis*? We have already noted the possibility that iron-dependent *trpBA* regulation in *C. trachomatis* may enable the induction of a similar response to both Trp and iron starvation, stimuli likely mediated by IFN-γ in vivo. This mechanism also presents the opportunity for *C. trachomatis* to respond similarly to distinct *sequential* stresses, where a particular stress may prime the pathogen to better cope with subsequent stresses. To reach the female upper genital tract (UGT), where most significant pathology is identified following infection with *C. trachomatis*, the pathogen must first navigate the lower genital tract (LGT). *Chlamydia* infections of the female LGT are associated with bacterial vaginosis (BV), which is characterized by obligate and facultative anaerobe colonization, some of which produce indole (*Sasaki-Imamura et al., 2011*; *Ziklo et al., 2016*). This provides *C. trachomatis* with the necessary substrate to salvage tryptophan via TrpBA. Interestingly, the LGT is also likely an iron-limited environment. Pathogen colonization and BV both increase the concentration of mucosal lactoferrin (Lf), an iron-binding glycoprotein, which can starve pathogens for iron (*Spear et al., 2011*; *Valenti et al., 2018*). Lf expression is additionally estrogen-regulated, and thus the LGT may normally experience periods of iron limitation (*Cohen et al., 1987*; *Kelver et al., 1996*). Intriguingly, *trpB* expression has been shown to be uniquely up-regulated in estradiol-supplemented infected cell cultures, perhaps indicating the involvement of estrogen-regulated mechanisms of cell-intrinsic iron starvation (*Amirshahi et al., 2011*). Moreover, the expression of TfR is constrained to the basal cells of the LGT stratified squamous epithelium (*Lloyd et al., 1984*), which likely restricts necessary Tf-bound iron from *C. trachomatis* infecting the accessible upper layers of the stratified epithelia (*Nogueira et al., 2017*; *Ouellette and Carabeo, 2010*).

For *C. trachomatis*, iron limitation may therefore serve as a critical signal in the LGT, inducing the expression of *trpBA* such that Trp is stockpiled from available indole, allowing the pathogen to counteract impending IFN-γ-mediated Trp starvation. We suggest the possibility that iron limitation in the LGT may be a significant predictor of successful pathogen colonization in the UGT and that iron-dependent regulation of *trpBA* may be an important virulence trait in genital serovars of *C. trachomatis*. Unfortunately, testing these hypotheses in cell culture models of infection presents a significant challenge. Evaluating rescue of chlamydial growth in the presence of indole to specifically assess the iron-dependent role of *trpBA* requires simultaneous Trp and iron depletion. The former ensures indole utilization by the bacteria, and the latter de-represses YtgR-regulated *trpBA* expression. In theory, this is feasible, but in practice the combined stress rapidly induces aberrant development, muddying results obtained from such studies (data not shown). Ideally, genetic approaches could be employed to distinguish the regulatory effects of YtgR independent of TrpR. However, the genetic manipulation of *trans*-acting factors (*e.g.* YtgR) will presumably have unpredictable off-target effects. Genetically altering *cis*-acting factors – such as operator sequences – is more feasible, but at

present we lack the information necessary to rationally mutate these sequences in *C. trachomatis* to interrogate these questions. The tight regulatory coordination at both the transcription initiation and termination steps would likely mean any mutation in the *cis*-acting sequences would affect both processes indiscriminately. Furthermore, in vivo infection models present challenges: attempting to answer these questions will likely require the use of non-human primate studies, as mouse models of *Chlamydia* infection do not recapitulate immune-mediated Trp starvation (*Nelson et al., 2005*). Ultimately, these limitations do not undermine the biological significance of an iron-dependent mode of regulating Trp salvage, given the critical role played by this pathway during infection.

Finally, and of note, the expression of the unique class Ic ribonucleotide diphosphate reductase-encoding *nrdAB* was also recently shown to be iron-regulated in *C. trachomatis* (*Brinkworth et al., 2018*). The regulation of *nrdAB* is known to be mediated by the presumably deoxyribonucleotide-dependent transcriptional repressor NrdR, encoded distal to the *nrdAB* locus (*Case et al., 2011*). As NrdR activity is not known to be modulated by iron availability, this raises the intriguing possibility that here too a unique iron-dependent mechanism of regulation may integrate chlamydial stress adaptations to promote a unified response across various stresses. Future studies may require more metabolomics-based approaches to thoroughly dissect the integration of these stress responses, as transcriptome analyses alone often miss broader, pathway-oriented metabolic coordination. Ultimately, these studies point towards a need to carefully re-evaluate the molecular stress response in *Chlamydia*, with greater emphasis on the use of targeted approaches and treatment protocols that induce stress, but not persistence. We anticipate that the rapid progress of the field in recent years will continue to catalyze exciting and important discoveries regarding the fundamental biology of *Chlamydia*.

# Materials and methods

### Key resources table

| Reagent type (species) or resource | Designation | Source or reference | Identifiers | Additional information |
|---|---|---|---|---|
| Gene (*Chlamydia trachomatis*) | YtgR | *Thompson et al., 2012* | CTL0325 | C-terminal 139 amino acids of YtgC |
| Strain, strain background (*Chlamydia trachomatis*) | L2 434/Bu | other | NC_010287 | No RRID |
| Strain, strain background (*Escherichia coli*) | BL21(DE3) | Sigma-Aldrich | CMC0016 | Electrocompetent cells |
| Cell line (*Homo sapiens*) | HeLa 229 | ATCC | RRID:CVCL_1276 | cervical adenocarcinoma epithelial cells |
| Antibody | His-Tag (D3I1O) XP | Cell Signaling Technology | RRID:AB_2744546 | Conditions used are described in Materials and methods |
| Recombinant DNA reagent | | | | |
| Sequence-based reagent | pCCT101 | *Thompson et al., 2012* | | Reporter gene plasmid for lacZ two-plasmid assay |
| Sequence-based reagent | pET151/D-TOPO | Invitrogen | K15101 | |
| Commercial assay or kit | RiboPure RNA Purification Kit, bacteria | Invitrogen | AM1925 | Modifications to manufacturer protocol described in Materials and methods |

*Continued on next page*

*Continued*

| Reagent type (species) or resource | Designation | Source or reference | Identifiers | Additional information |
|---|---|---|---|---|
| Commercial assay or kit | SMARTer RACE 5′/3′ Kit | Takara Bio | 634859 | Modifications to manufacturer protocol described in Materials and methods |
| Chemical compound, drug | 2,2-bipyridyl (Bpdl) | Sigma-Aldrich | D216305 | Prepared at 100 mM in 100% Ethanol; used at 100 µM working concentration |
| Software, algorithm | R Studio | http://www.rstudio.com/ | | RStudio Team (2016). RStudio: Integrated Development for R. RStudio, Inc, Boston, MA |

## Eukaryotic cell culture and chlamydial infections

Human cervical epithelial adenocarcinoma HeLa (ATCC CCL-2; Purchased 08/2016; Last tested for *Mycoplasma* 07/2018) cells were cultured at 37˚ C with 5% atmospheric $CO_2$ in Dulbecco's Modified Eagle Medium (DMEM) supplemented with 10 µg/mL gentamicin, 2 mM L-glutamine, and 10% (*v/v*) filter sterilized fetal bovine serum (FBS). For all experiments, HeLa cells were cultured between passage numbers 4 and 16. *Chlamydia trachomatis* serovar L2 (434/Bu) was originally obtained from Dr. Ted Hackstadt (Rocky Mountain National Laboratory, NIAID). Chlamydial EBs were isolated from infected HeLa cells at 36–40 hr post-infection (hpi) and purified by density gradient centrifugation essentially as described (*Caldwell et al., 1981*).

For the infection of 6-well tissue culture plates, HeLa cells cultured to 80–90% confluency were first washed with pre-warmed Hanks Buffered Saline Solution (HBSS) prior to the monolayer being overlaid with inoculum (un-supplemented DMEM) at the indicated multiplicity of infection (MOI). Tissue culture plates were then centrifuged at 4˚ C with a speed of 1000 RPM (Eppendorf 5810 R table top centrifuge, A-4–81 rotor) for 5 min to synchronize the infection. Inoculum was aspirated and cells were washed again with pre-warmed HBSS prior to the media being replaced with pre-warmed complete DMEM. Infected cultures were then returned to the tissue culture incubator until the indicated times post-infection. This procedure was replicated exactly for the infection of 24-well tissue culture plates.

## Iron starvation

*Chlamydia trachomatis* L2-infected HeLa cell cultures were starved for iron by supplementation of the media with the iron chelator 2,2-bipyridyl (Bpdl; Sigma Aldrich, St. Louis, MO, USA; CAS: 366-18-7) essentially as described (*Thompson and Carabeo, 2011*). Briefly, at the indicated times post-infection, infected cell cultures were washed with pre-warmed HBSS prior to the addition of complete DMEM (mock) or complete DMEM supplemented with 100 µM Bpdl. Infected cell cultures were returned to the incubator for the indicated treatment periods. Bpdl was prepared as a 100 mM stock solution in 100% ethanol and stored at −20˚ C for no longer than 6 months.

## Tryptophan starvation

*Chlamydia trachomatis* L2-infected HeLa cell cultures were starved for tryptophan by replacement of complete DMEM with tryptophan-depleted medium. In brief, Tryptophan-replete or –deplete DMEM-F12 (U.S. Biological Life Sciences, Salem, MA, USA) powder media was prepared following manufacture instructions and supplemented with 10% (*v/v*) filter-sterilized FBS which had been previously dialyzed 16–20 hr at 4˚ C in PBS in a 10 kDa MWCO dialysis cassette. Media was then further supplemented with 10 µg/mL gentamicin. At the indicated times post-infection, complete DMEM was aspirated and wells were washed with pre-warmed HBSS prior to the addition of tryptophan-

replete or –deplete medium. Infected cell cultures were returned to the incubator for the indicated treatment periods.

## Cloning

All constructs were cloned using standard molecular cloning techniques, *e.g.* restriction enzyme, homology-directed, etc. All primers and plasmids used in this study can be found in *Supplementary file 5* and *6*, respectively. All pCCT constructs were cloned by amplifying the promoter region of interest with 5' and 3' flanking KpnI sites, which were then KpnI digested (New England Biolabs, Ipswich, MA, USA) along with the pCCT-*ytgA* vector (to excise the *ytgA* promoter fragment). The vector was then treated with antarctic phosphatase (New England Biolabs) prior to having the promoter of interest ligated into the pCCT backbone. Inserted promoters were verified to be in the correct orientation and free of sequence errors by PCR and sequencing. All pET vectors were cloned following manufacturer instructions (Invitrogen, ThermoFisher Scientific, Waltham, MA, USA).

## Immunofluorescent confocal microscopy

At the indicated times post-infection, *C. trachomatis* L2-infected HeLa cell cultures seeded on glass coverslips in 24-well tissue cultures plates were first washed with pre-warmed HBSS prior to fixation with 4% paraformaldehyde (PFA) in phosphate buffered saline (PBS) for 20 min at RT ° C. Fixation solution was aspirated and wells were washed with PBS prior to permeabilization with 0.2% Triton X-100 in PBS for 5 min at RT° C. Permeabilization solution was then decanted and cells were washed with PBS. The coverslips were blocked for 30 min with 1% bovine serum albumin (BSA) in PBS at RT° C. To stain for *Chlamydia*, coverslips were washed with PBS and PBS supplemented with 1% BSA and 1:500 convalescent human sera was added to wells and incubated at RT° C for 1 hr with rocking. Primary antibody solution was decanted and coverslips were again washed with PBS. Goat anti-human Alexa-647 (Invitrogen, ThermoFisher Scientific) diluted 1:1000 in PBS with 1% BSA was then added to the wells and incubated in the dark for another hour at RT° C with rocking. Secondary antibody solution was then decanted, coverslips were washed again with PBS and coverslips were either immediately mounted on microscopy slides using ImmuMount (ThermoFisher Scientific) or VectaShield H-1000 (Vector Laboratories, Burlingame, CA, USA) or stored in the dark at 4° C until mounting. All images were acquired on a Leica TCS SP8 laser scanning confocal microscope, using identical settings, in the Integrative Physiology and Neuroscience Advanced Imaging Center at Washington State University. All images are Z-projections and were processed in Fiji (*Schindelin et al., 2012*) and Adobe Creative Suite identically for each comparative time-point.

## Nucleic acid preparation

RNA was harvested from *C. trachomatis*-infected HeLa cell monolayers by scraping 3 wells of a 6-well plate in ice-cold Trizol Reagent (ThermoFisher Scientific). Samples were then pooled and split into two technical replicates (RT-qPCR) or kept as one biological replicate (RACE). Trizol-extracted samples were then thoroughly vortexed with a 100 µL volume of Zirconia beads prior to chloroform extraction. 100% ethanol was added to the aqueous phase and RNA was isolated using the Ambion RiboPure RNA Purification kit for bacteria following manufacturer instructions (ThermoFisher Scientific). DNA was removed from RNA samples using the Invitrogen DNA-*free* DNA Removal Kit following manufacturer instructions (ThermoFisher Scientific). RNA was stored at −20° C until further use. For *E. coli*, RNA was harvested using the Ambion RiboPure RNA Purification kit for bacteria following manufacturer instructions (ThermoFisher Scientific) from 9 mL of bacterial culture prepared as described below for the Two-Plasmid Reporter Assay. RNA was subsequently DNased as described above. cDNA was generated using either SuperScript IV Reverse Transcriptase (RT-qPCR; ThermoFisher Scientific) or SMARTScribe Reverse Transcriptase (RACE and RACE-specific qRT-PCR); Takara Bio, Kusatsu, Shiga Prefecture, Japan) essentially as described by the respective manufacturers. For cDNA generated for RT-qPCR, 650 ng of total RNA was used as a template in a 20 µL total reaction volume. For every RT reaction, a 'no-RT' control, generated from 350 ng of total RNA template in a 10 µL total volume, was included. For 5'-RACE, cDNA was generated from 250 ng of total RNA using random primers in a 10 µL total volume and further processed in the RACE workflow. cDNA was stored at −20° C.

gDNA was harvested from *C. trachomatis*-infected HeLa cell monolayers by scraping 3 wells of a 6-well plate in ice-cold PBS + 10% Proteinase K (ThermoFisher Scientific). Samples were then pooled and split into two technical replicates for analysis of genome copy number by qPCR. gDNA was isolated using the DNeasy Blood and Tissue Kit following manufacturer protocols (QIAGEN, Hilden, Germany). gDNA was stored at $-20°$ C until further use.

Reverse Transcription Quantitative Polymerase Chain Reaction (RT-qPCR) cDNA (or gDNA in qPCR), prepared as described above, was diluted 1:10 or 1:100 in nuclease-free $H_2O$ depending on the experimental condition being assayed (*e.g.* treatment, point in development cycle, etc.). On ice, 3.3 µL of diluted sample was added to 79 µL of PowerUp SYBR Green Master Mix (ThermoFisher Scientific) with specific qPCR primers diluted to 500 nM. From this master mix, each experimental sample was assayed in triplicate 25 µL reactions. Assays were run on an Applied Biosystems 7300 Real Time PCR System with cycling conditions as follows: Stage 1: 50.0° C for 2 min, one rep. Stage 2: 95.0° C for 10 min, one rep. Stage 3: 95.0° C for 15 s, 40 reps. Stage 4: 60.0° C for 1 min, one rep. Primers were subjected to dissociation curve analysis to ensure that a single product was generated. For each primer set, a standard curve was generated using purified *C. trachomatis* L2 gDNA from EB preparations diluted from $2 \times 10^{-3}$ to $2 \times 10^0$ ng per reaction. $C_t$ values generated from each experimental reaction were then fit to standard curves (satisfying an efficiency of $95 \pm 5\%$) for the respective primer pair and from the calculated ng quantities, transcript or genome copy number was calculated as follows:

$$Genome\ copy\ number\left(\frac{genome\ copies}{ng\ total\ gDNA}\right) = \frac{ng\ genome\ \times\ df}{ng\ total\ gDNA} \times \frac{892,000\ copies}{ng\ DNA}$$

$$Transcript\ copy\ number\left(\frac{transcript\ copies}{\frac{genome\ copies}{ng\ total\ gDNA}}\right) = \frac{ng\ transcript\ \times\ df}{\frac{genome\ copies}{ng\ total\ gDNA}} \times \frac{892,000\ copies}{ng\ DNA}$$

Where *df* = dilution factor and the number of copies/ng DNA is calculated based on the size of the *C. trachomatis* L2 genome assuming that the molar mass per base pair is 650 (g/mol)/bp (note that this value should be the same for any single-copy ORF on the genome). All quantifications of genome copy number were determined using the *ahpC* qPCR primer set. Values from replicate assays were averaged, and values from replicate RNA/gDNA isolations were averaged to obtain the mean and standard deviation for one biological replicate. For some experiments, to account for batch effects across biological replicates, data was transformed such that the mean of all samples in each replicate was identical. In some instances, batch correcting generated negative values, and in this case data sets were scaled such that the lowest value equaled 1.0. HeLa cells were infected at an MOI of 2 for all RT-qPCR studies.For analysis of transcriptional readthrough, RT-qPCR was performed as described above and the readthrough value was computed as:

$$Readthrough = 2^{\left(C_{t(Normalization)} - C_{t(Experimental)}\right)}$$

Readthrough values were then batch corrected such that the mean of each replicate was identical.

## 5' Rapid Amplification of cDNA Ends (5'-RACE)

All RACE studies were performed using the SMARTer RACE 5'/3' Kit (Takara Bio). To observe 5'-RACE products from the *trpRBA* operon, a 'nested' RACE protocol was used as outlined in the SMARTer RACE 5'/3' Kit user manual. Briefly, 1.25–2.5 µL of cDNA generated for RACE was added to a 25 µL reaction volume and run in a thermal cycler for 40 cycles using the touch-down PCR conditions described by the manufacturer. In brief, five cycles were run at an annealing temperature of both 72° C and 70° C prior to 30 cycles run with an annealing temperature of 68° C. Following this primary amplification, the RACE products were diluted 1:50 in Tricine-EDTA Buffer supplied by the manufacturer, and 2.5 µL of diluted primary RACE product was added to a 25 µL reaction volume and subjected to another 20 cycles of nested PCR, as described by the manufacturer, using primers designed within the amplicon of the primary RACE products. Samples were electrophoresed on a 2% agarose gel for visualization and analysis. HeLa cells were infected at a MOI of 5 for all RACE studies.

## 3' Rapid Amplification of cDNA Ends (3'-RACE)

3'-RACE studies were performed essentially identical to 5'-RACE with the exception that total RNA was subjected to poly(A) tailing with a Poly(A) Polymerase following manufacturer instructions (New England Biolabs). In brief, at least 3.5 µg of total RNA was incubated at 37° C with Poly(A) Polymerase in reaction buffer supplemented with ATP and murine RNase Inhibitor (New England Biolabs) for 30 min prior to heat-inactivation at 65° C for 20 min. RNA was re-isolated through an RNA clean-up filter cartridge (Ambion, ThermoFisher Scientific). A total of 125 ng of poly(A)-tailed total RNA was then used to generate 3'-RACE ready cDNA in a 10 µL reaction volume following manufacturer instructions. Primary and nested RACE was performed using 3'-RACE gene-specific primers following the same protocol for amplification described for 5'-RACE, with the exception that the extension time was adjusted to accommodate amplification of the full ~3 kb *trpRBA* polycistronic message.

## Mapping of 5'/3'-RACE Products

5'-RACE products generated from either primary or nested RACE reactions were excised from the agarose gel and DNA was isolated using the NucleoSpin Gel and PCR Clean-up kit (Macherey-Nagel, Takara Bio). The isolated RACE products were then cloned into the pRACE vector supplied in the SMARTer RACE 5'/3' Kit using the In-Fusion HD cloning kit (Takara Bio). Ligated vectors were transformed into chemically competent Stellar *E. coli* cells by heat shock. Transformed bacteria were plated on LB agar containing 50 µg/mL carbenicillin and incubated overnight at 37° C. Colonies were selected and screened for relevant inserts by PCR. Positive colonies were cultured overnight at 37° C in LB liquid broth containing 50 µg/mL carbenicillin and plasmids were isolated using the QIAprep Spin Miniprep kit (QIAGEN). Inserts were then sequenced by Eurofins Genomics using the default M13 Reverse sequencing primer. Returned sequencing data was aligned to the *C. trachomatis* L2 (434/Bu) genome (NCBI Accession: NC_010287) by BLAST and the most 5' aligned nucleotide was considered the 5' end of the insert. In the case of 3'-RACE data, the reverse complement sequence was first generated prior to alignment. Grouping of individual products was determined 1.) by clusters being greater than 30 nucleotides apart and 2.) by the specific RACE band that the alignment was derived from. These two criteria were not both satisfied in all cases and in those cases criteria 1.) was favored.

## Sequence alignments

All *C. trachomatis* L2 434/Bu genome sequences were obtained from NCBI Accession NC_010287. Global pairwise sequence alignments were made using the EMBOSS Needle algorithm. Alignment parameters were set as follows: Matrix: DNAfull, Gap Open: 20, Gap Extend: 0.8, Output Format: pair, End Gap Penalty: True, End Gap Open: 10, End Gap Extend: 0.5. These conditions were sufficient to replicate the previously published alignment between the putative YtgR operator sequence and the TroR operator (*Akers et al., 2011*). Local pairwise sequence alignments were made using the EMBOSS Water algorithm. The putative YtgR operator was aligned to the entire 348 bp intergenic region of the *trpRBA* operon (*C. trachomatis* L2 [434/Bu] genome position 511,692–512,039). The alignment parameters were set as follows: Matrix: DNAfull, Gap Open:10, Gap Extend: 0.5, Output Format: pair. These are the default conditions and were chosen to remove bias from the alignment results.

## Two-Plasmid reporter assay

The YtgR-binding reporter assay was performed essentially as described, with minor modifications (*Thompson et al., 2012*). Promoter regions of interest were amplified from the *C. trachomatis* L2 (434/Bu) genome by PCR using the indicated primer sets, which included KpnI restriction endonuclease sites at the 5' and 3' ends of the promoter amplicon. The amplified fragments and the pCCT-EV plasmid were then KpnI-digested and the promoters ligated into the vector using T4 or Quick Ligase (New England BioLabs). Insert directionality was confirmed by directional colony PCR and positive clones were sequence verified. pCCT-*trpBA*ΔOperator was cloned by amplifying two fragments of the pCCT-*trpBA* vector with one ~ 60 mer primer containing the bases to be substituted for each fragment. Thus, the whole vector was split into two half-fragments containing the substituted bases. The two fragments were then cloned back together using In-Fusion Homology-Directed cloning (Takara Bio) to yield the final vector. Electrocompetent BL21(DE3) *E. coli* (Sigma Aldrich) were co-

transformed by electroporation with the pCCT reporter plasmid and the pET151 expression vector (-EV or –YtgR) and plated on double selective LB agar containing 50 µg/mL carbenicillin and 15 µg/mL tetracycline. Prior to plating of transformed cells, 50 µL of 40 mg/mL X-Gal in DMSO (EMD Millipore, Burlington, MA, USA) was applied to the plate for colorimetric determination of β-galactosidase expression. Transformants were incubated overnight at 37° C. The following evening, blue colonies from each experimental condition were selected and cultured overnight in LB liquid broth containing 0.2% (w/v) D-glucose (for catabolite repression of expression vectors), 50 µg/mL carbenicillin and 15 µg/mL tetracycline. Cultures were incubated overnight at 37° C. The following morning, overnight cultures were spun down to remove glucose-containing media and sub-cultured in LB liquid broth medium containing 50 µM $FeSO_4$, 50 µg/mL carbenicillin and 15 µg/mL tetracycline to an $OD_{600}$ of 0.45. Cultures were incubated for 1 hr at 37° C and sub-cultured a second time in the same media to an $OD_{600}$ of 0.1. Cultures were returned to the incubator for another hour prior to the addition of 500 µM isopropyl β-D-1-thiogalactopyranoside (IPTG) to induce pET151 expression from the *lac* promoter. Cultures were incubated another hour prior to the addition of 0.2% L-arabinose to induce *lacZ* expression from the *araBAD* promoter. Cultures were incubated a final 2 hr prior to the collection of a 0.1 mL volume of cells for assaying β-galactosidase activity by the Miller Assay (*Miller, 1972*). Cell pellets were stored at −80° C prior to being assayed. To assay β-galactosidase activity, cell pellets were first re-suspended in Z-buffer (pH 7.0, 60 mM $Na_2HPO_4$, 40 mM $NaH_2PO_4$, 10 mM KCl, 1 mM $MgSO_4$ and 2.7 µL/mL β-mercaptoethanol). 50 µL of 0.1% SDS and 100 µL of chloroform were then added to each sample prior to thorough vortexing. Samples were equilibrated for 5 min at 30° C and 200 µL of 4 mg/mL ortho-nitrophenyl-β-galactoside (ONPG) prepared in Phosphate Buffer (pH 7.0, 60 mM $Na_2HPO_4$, 40 mM $NaH_2PO_4$) were added to the samples to initiate the reaction. Reactions were stopped by the addition of 500 µL 1 M $Na_2CO_3$. Absorbance was measured on a FLUOStar Optima plate reader (BMG Labtech, Offenburg, Germany) at 420 nm and Miller Units were calculated as:

$$1000 \times \frac{Abs_{420}}{t \times v \times OD_{final}}$$

Where $t$ = reaction time, $v$ = volume of cells and $OD_{final}$ = $OD_{600}$ at the time of sample collection. It was empirically determined that the subtraction of absorbance at 550 nm had a negligible effect on the calculated value. A blank sample lacking cells was included in each experimental batch and used as a reference for absorbance. For each experimental condition, three independent co-transformed colonies were assayed in technical triplicate. In some instances, significantly high Miller Unit outliers were excluded by Grubb's Test (p<0.05) under the assumption that extreme *lacZ* expression may reflect plasmid copy number or reporter gene expression issues.

## Two-Plasmid chromatin immunoprecipitation (ChIP)
### Preparation of lysates
Co-transformed BL21(DE3) *E. coli* were cultured exactly as described for the Two-Plasmid assay with the exception that volumes were scaled up proportionately to increase sample size. For bipyridyl-treated experiments, 50 µM $FeSO_4$ was excluded and 500 µM 2,2-bipyridyl in DMSO was added for the final three hours of culturing alongside IPTG. At the end of culturing, samples were fixed by diluting fresh 37% methanol-stabilized formaldehyde solution (Sigma Aldrich) to a final concentration of 1% and incubating the sample for 10 mins with periodic vortexing. Fixation was then quenched by the addition of 250 mM glycine to each sample and incubation proceeded for another five mins with periodic vortexing. Samples were then centrifuged for 15 mins at 3000xg, 4 °C to pellet fixed cells. The supernatant was discarded and samples were stored at −80° C. Fixed samples were thawed on ice and resuspended in 1 mL of ice-cold Pierce RIPA Lysis Buffer (ThermoFisher Scientific) supplemented with cOmplete EDTA-free Protease Inhibitor cocktail (Roche, Basel, Switzerland). Samples were then divided into three 330 µL aliquots (experimental IP, control IP and input) and immediately processed for sonication. Sonication was performed using the Biorupter Plus sonication device (Diagenode, Denville, New Jersey, USA) with an attached water cooler. Samples were sonicated on high power for a total of 20 thirty-second on/off cycles at 4° C. These sonication conditions were not optimized for a particular DNA fragment size given that the target DNA promoter element was located on a plasmid and therefore isolated from possible off-target genomic binding sites.

Sonicated lysates were then centrifuged to clear cell debris at 14,800 RPM for 5 mins at 4° C in a Sorvall Legend Mirco 21R centrifuge (ThermoFisher Scientific). Lysates were pre-cleared by incubating with 10 µL of Pierce Protein A/G Magnetic beads (ThermoFisher Scientific) at 4° C with rotation for 1 hr. Cleared lysates were then immediately processed for immunoprecipitation.

## Immunoprecipitation

For experimental IPs, Pierce Protein A/G Magnetic beads were loaded with His-Tag (D3I1O) XP Rabbit monoclonal antibody (12698, Cell Signaling Technology, Inc, Danvers, MA, USA). Control IPs were performed with MNormal Rabbit IgG, (2729S, Cell Signaling Technology, Inc). Antibody-loaded beads were prepared by incubating 25 µL of beads with 10 µL of antibody in 500 µL TBS-T (ThermoFisher Scientific) at 4° C for 2.5 hr with rotation. Antibody beads were separated on a magnetic rack and then blocked in 500 µL 5% BSA +200 mg/mL sheared salmon sperm DNA (Invitrogen) by incubating for 1 hr at 4° C with rotation. Blocked antibody beads were separated on a magnetic rack and then resuspended in TBS-T to their original volume. These conditions were scaled proportionately when doing multiple IPs. 25 µL of blocked antibody beads were then added to 330 µL of cleared sonicated lysate and incubated overnight at 4° C with rotation. The following morning, beads were separated on a magnetic rack and washed first with 300 µL ice-cold Low Salt Immune Complex Wash Buffer (EMD Millipore) incubated for five mins at RT° C with rotation. Beads were separated on a magnetic rack and washed a second time with 300 µL ice-cold High Salt Immune Complex Wash Buffer (EMD Millipore) incubated for five mins at RT° C with rotation. A final wash was carried out using 300 µL ice-cold LiCl Immune Complex Wash Buffer (EMD Millipore) incubated for five mins at RT° C with rotation. Protein-DNA complexes were then eluted off of the washed beads by incubating the beads in 420 µL Pierce IgG Elution Buffer, pH 2.0 (ThermoFisher Scientific) for five mins at RT° C with rotation. The beads were then immediately separated on a magnetic rack and the eluate was added to 80 µL 1 M Tris, pH 8.5 to neutralize pH.

## DNA isolation

Cross-links were reversed by incubating eluted Protein-DNA complexes and input lysates at 95° C for 1 hr, with 750 RPM shaking. De-crosslinked samples were then centrifuged at 14,800 RPM for 5 mins at 4° C to clear debris. DNA was isolated using the Macherey-Nagel NucleoSpin Gel and PCR Clean-up Kit as described by the manufacturer (TakaraBio) with the stipulation that the input lysates were bound to the column in Buffer NTB and the immunoprecipitated DNA was bound to the column in Buffer NTI. DNA was eluted in a total of 30–60 µL Buffer NE and then processed for qPCR using the indicated primer pairs. ChIP-Input was diluted 1:50 or 1:100 in nuclease-free water prior to qPCR.

## Fold enrichment quantitation

The $C_t$ values obtained from triplicate qPCR assays were averaged and dilution correction was calculated as:

$$C_{t(DF)} = C_t - Log_2(DF)$$

Where dilution factor equals the fold-dilution reported above (*e.g.* 50).
The percent recovery from ChIP-Input was calculated as:

$$\%Recovery = 100 \times 2^{C_{t(Input)} - C_{t(ChIP)}}$$

Where $C_{t(Input)}$ represents the $C_t$ value of the dilution-corrected ChIP-Input sample and $C_{t(ChIP)}$ represents the dilution-corrected $C_t$ value of the experimental or control antibody IPs.
The fold enrichment relative to the control antibody IP was calculated as:

$$Fold\,Enrichment = \frac{\%\ Recovery\ Experimental\ ChIP}{\%\ Recovery\ Control\ ChIP}$$

These calculations assume 100% qPCR efficiency.
The Fold Enrichment value was then normalized to the copy number of pET151-YtgR to account for variation in YtgR expression and thus ChIP efficacy. Copy number for pET151-YtgR was

determined using primer pairs specific for the ampicillin resistance cassette and copy number was calculated as described above based on the size of the pET151-YtgR vector. Experimental values obtained from these assays were then batch corrected such that the mean of all samples in each replicate was identical. For aesthetic reasons, the values were then scaled such that the lowest value equaled 1.0.

## RNA-Sequencing

RNA-Sequencing experiments were performed as described in their original publication (*Brinkworth et al., 2018*). Raw and processed sequencing files were submitted to the NCBI Gene Expression Omnibus (GEO) as a Superseries and can be found at accession number GSE106763. Coverage maps were generated by mapping all reads across three biological replicates to a single reference file in CLC Genomics Workbench v11. To facilitate easy analysis of IGR boundaries, the *C. trachomatis* L2 434/Bu genome (Accession: NC_010287) was modified to contain annotations for intergenic regions that fell between two genes in the same coding orientation, and this genome was used as the reference for read mapping. Read mapping and differential expression analysis was performed using default settings in CLC Genomics Workbench. Data aggregation in the Reads track was set to aggregate above 1 bp.

## Graphs and statistical analysis

All graphs were generated using the ggplot2 package (*Wickham, 2009*) in R Studio, and/or in the Adobe Creative Suite. All line plots and bar graphs represent the mean $\pm$ one standard deviation unless otherwise noted. All box and whisker plots represent the distribution of data between the $1^{st}$ and $3^{rd}$ quartile range within the box, while the whiskers represent data within 1.5 interquartile ranges of the $1^{st}$ or $3^{rd}$ quartile. Extreme values outside this range are plotted as open circles. The $2^{nd}$ quartile (median) is plotted as a black line within the box. Histogram plots were generated with a bin width of 20 and are plotted on a density scale. The overlaid density plots represent a statistical approximation of the data over a continuous scale. All statistical analyses were carried out in R Studio. All statistical computations were performed on the mean values of independent biological replicates calculated from the indicated number of respective technical replicates. For single pairwise comparisons, a two-sided unpaired Student's *t*-test with Welch's correction for unequal variance was used to determine statistical significance. For multiple pairwise comparisons, a One-Way Analysis of Variance (ANOVA) was conducted to identify significant differences within groups. If a significant difference was detected, then the indicated post-hoc pairwise test was used to identify the location of specific statistical differences. A *p*-value less than 0.05 was considered statistically significant. For all figures, *=$p < 0.05$, **=$p < 0.01$, and ***=$p < 0.005$.

## Acknowledgements

We thank Liam Caven, Korinn Murphy and Matthew Romero for critical review of this manuscript; Dr. Christopher C Thompson for the establishment of the *E. coli* YtgR reporter system and generation of the pCCT construct; Dr. Scot P Ouellette for support, critical feedback and advice; and Dr. David Dewitt for expert advice, training and maintenance of equipment in the IPN Advanced Imaging Center. This work was supported by NIH grants R01-AI065545 and R01-AI132406 to RAC; F31-AI136295 and T32-GM008336 to NDP; NDP was also supported by an Achievement Reward for College Scientists (ARCS; Seattle Chapter) Fellowship.

## Additional information

### Funding

| Funder | Grant reference number | Author |
| --- | --- | --- |
| National Institutes of Health | AI065545 | Rey Carabeo |
| Achievement Rewards for College Scientists Foundation | | Nick D Pokorzynski |
| National Institutes of Health | F31AI136295 | Nick D Pokorzynski |

| National Institutes of Health | T32GM008336 | Nick D Pokorzynski |
| National Institutes of Health | T32AI007025 | Amanda J Brinkworth |
| National Institutes of Health | AI132406 | Rey Carabeo |

The funders had no role in study design, data collection and interpretation, or the decision to submit the work for publication.

## Author contributions

Nick D Pokorzynski, Conceptualization, Resources, Data curation, Formal analysis, Funding acquisition, Validation, Investigation, Visualization, Methodology, Writing—original draft, Writing—review and editing; Amanda J Brinkworth, Data curation, Validation, Investigation, Methodology; Rey Carabeo, Conceptualization, Formal analysis, Supervision, Funding acquisition, Investigation, Writing—original draft, Project administration, Writing—review and editing

## Author ORCIDs

Nick D Pokorzynski (iD) http://orcid.org/0000-0003-2438-2368
Amanda J Brinkworth (iD) http://orcid.org/0000-0003-3340-8494
Rey Carabeo (iD) http://orcid.org/0000-0002-5708-5493

## Decision letter and Author response

Decision letter https://doi.org/10.7554/eLife.42295.041
Author response https://doi.org/10.7554/eLife.42295.042

# Additional files

## Supplementary files

• Source code 1. Template R code for generation of histogram with overlaid density plot as depicted in *Figures 4C* and *6C*.
DOI: https://doi.org/10.7554/eLife.42295.028

• Source code 2. Template R code for computation of One-Way ANOVA with post-hoc pairwise t-tests.
DOI: https://doi.org/10.7554/eLife.42295.029

• Source code 3. Template R code for computation of two-sample t-test with Welch's correction for unequal variance.
DOI: https://doi.org/10.7554/eLife.42295.030

• Supplementary file 1. 5'-RACE BLAST dataset.
DOI: https://doi.org/10.7554/eLife.42295.031

• Supplementary file 2. 3'-RACE BLAST dataset.
DOI: https://doi.org/10.7554/eLife.42295.032

• Supplementary file 3. 5'-RACE Sequence dataset.
DOI: https://doi.org/10.7554/eLife.42295.033

• Supplementary file 4. 3'-RACE Sequence dataset.
DOI: https://doi.org/10.7554/eLife.42295.034

• Supplementary file 5. Primers used in this study.
DOI: https://doi.org/10.7554/eLife.42295.035

• Supplementary file 6. Plasmids used in this study.
DOI: https://doi.org/10.7554/eLife.42295.036

• Transparent reporting form
DOI: https://doi.org/10.7554/eLife.42295.037

## Data availability

Raw and processed sequencing data have been deposited in the Gene Expression Omnibus (Accession number GSE106763).

The following previously published dataset was used:

| Author(s) | Year | Dataset title | Dataset URL | Database and Identifier |
|---|---|---|---|---|
| Brinkworth AJ, Wildung MR | 2018 | Transcriptional response of C. trachomatis to early-cycle and mid-cycle iron-starvation | https://www.ncbi.nlm. nih.gov/geo/query/acc. cgi?acc=GSE106763 | Gene Expression Omnibus, GSE106763 |

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
