## [Decision Letter]

Thank you for submitting your article "A bipartite iron-dependent transcriptional regulation of the tryptophan salvage pathway in *Chlamydia trachomatis*" for consideration by *eLife*. Your article has been reviewed by 3 peer reviewers, and the evaluation has been overseen by Gisela Storz as the Reviewing and Senior Editor. The following individual involved in review of your submission has agreed to reveal their identity: Derek Fisher (Reviewer #2).

The reviewers have discussed the reviews with one another and the Reviewing Editor has drafted this decision to help you prepare a revised submission.

Summary:

This study investigates iron-regulated expression of *C. trachomatis* tryptophan biosynthetic genes, which have been linked to surviving IFN-Ɣ stimulation of host cells and thus virulence. The authors established experimental conditions to decouple the effects of iron and trp starvation, both of which can be induced by IFN-Ɣ stimulation, and they showed discoordinate regulation of the *trpRBA* operon by iron. Iron regulation appears to be mediated through binding of the iron-dependent regulator, YtgR, to the trpR-trpB intergenic region (IGR) upstream of a newly-identified, iron-regulated promoter. The authors also explored the possibility that YtgR binding to the IGR causes transcriptional termination to reduce expression of *trpBA*, although the data for this model are less convincing.

Essential revisions:

1) The βgal results reported in Figure 5 and Figure 5—figure supplement 1 indicate that binding of YtgR to the IGS may require more than the putative operator sequence. Demonstration of iron-dependent, direct interactions between YtgR and the IGS/operator site would provide clarity on this point. EMSA or YtgR/operator DNA dot blot assays previously reported by the Carabeo group (PMID: 22689982) measuring iron-dependent, direct binding to the IGS region would provide strong support for the trp IGS YtgR operator site designation and iron-regulation, and could allow for rapid screening of sequence elements needed for YtgR binding. The latter approach might allow for the in silico identification of other YtgR operator sites.

2) The section on iron limitation and transcription read-through was not well written. It was hard to understand and not fully explained. The three graphs should be shown at the same scale so that they can be fairly compared. What is the significance of decreased read-through at 18 hpi compared to 12 hpi, and which should BpdI treatment be compared to? This is not a trivial consideration because the difference between 12 and 18 hpi could potentially be due to increased attenuation by tryptophan and/or iron levels. What does it mean that there was less read-through with depletion or iron than tryptophan? The presence of residual read-through despite iron depletion should be acknowledged and discussed. The result with tryptophan depletion was described as being expected but the reasoning was not given. How do you know if tryptophan and iron affect read-through through the same or different mechanisms?

3) It's critical for the study to establish iron-starvation conditions that don't induce persistence, but 1 of the 2 persistence markers did not respond as expected. Additional transcriptional markers of persistence (Belland, 2003) should be tested to confirm that 6 hours of BpdI treatment did not induce persistence. Ideally these persistence marker should include iron-dependent and iron-independent genes.

4) The RT-qPCR results in Figure 1D showing an increase in transcript levels of the late gene omcB following iron depletion are at odds with prior work from the group using a *C. trachomatis* serovar D isolate (Thompson and Carabeo, 2011) which showed decreased omcB transcript levels. This is touched on by the authors, who point out that omcB levels and iron stress responses seem to vary across chlamydial species and *C. trachomatis* serovars. This creates a potential concern that *trpBA* iron-mediated regulation might be LGV specific and may not occur in serovar D-K isolates which are more prevalent causes of infection in females (and the female reproductive tract in particular). This becomes problematic in relation to the hypothesis that YtgR-mediated regulation allows for differential responses to environments in the lower versus upper genital tract if serovars D-K do not show the LGV-type response. Alignment of the IGS regions across serovars might address this point. Experimentally, the βgal assay (Figure 5) using non-LGV IGS regions and confirmation of iron regulated control of *trpBA* (RT-qPCR, Figure 3) in non-LGV serovars would support that the results are not LGV-specific and occur in serovars more frequently isolated from infected women. This would increase the significance of the study.

5) While it is clear that trpR transcription terminates in the IGR, the data for iron-regulated termination within the IGR are fairly weak. For one, why were the comparisons in Figure 7C-E made to the 12hr instead of the 18hr time point, the latter of which appears to indicate a decrease in the ratio of both trpR and the IGR transcripts compared to read-through transcripts. Secondly, Figure 3C shows increased trpR levels in iron depleted condition when compared to mock treatment at the same infection time point, similar to the iron-regulated genes shown in Figure 2, indicating this transcript is regulated by iron. Without more in-depth analysis of trpR promoter activity and trpR transcript stability, I'm not certain these results can be interpreted as iron-regulation of transcriptional read through.

6) The impact of the study would be increased by data supporting the model that YtgR binding to its operator decreases RNA polymerase processivity. While these studies would be challenging in Chlamydia especially in a 2-3 month time frame expected for the revision, they could be done with the heterologous transcription assay. This approach would allow the effect of YtgR and other transcription factors on read-through transcription to be compared. It could also be used to show if this effect is dependent on iron or tryptophan. One concern regarding iron-regulated termination is that the data are not wholly consistent with this idea. Specifically, the data show some modest iron regulation of trpR, depending on what time points are used to evaluate the data (see Figures 3 and

7). Moreover, there are two TTS in the IGR downstream of the YtgR operator, suggesting termination can occur independently of YtgR binding. Thus it is possible that multiple factors, including RNA stability, RNA processing, and termination, are contributing to their observations. In the absence of additional data, the authors should temper their conclusions.

7) Is this the first demonstration of iron regulating a *C. trachomatis* virulence trait? This should be put into broader context of bacterial pathogenesis, where iron regulation of virulence is a recurrent theme for many pathogens, and highlighted as a new twist on this theme.

---

## [Author Response]

Essential revisions:1) The βgal results reported in Figure 5 and Figure 5—figure supplement 1 indicate that binding of YtgR to the IGS may require more than the putative operator sequence. Demonstration of iron-dependent, direct interactions between YtgR and the IGS/operator site would provide clarity on this point. EMSA or YtgR/operator DNA dot blot assays previously reported by the Carabeo group (PMID: 22689982) measuring iron-dependent, direct binding to the IGS region would provide strong support for the trp IGS YtgR operator site designation and iron-regulation, and could allow for rapid screening of sequence elements needed for YtgR binding. The latter approach might allow for the in silico identification of other YtgR operator sites.

To demonstrate iron-dependency of YtgR repression, we performed an additional two-plasmid experiment with the pET151-YtgR expression vector and the pCCT-*trpBA* reporter plasmid in the presence of 500 µM bipyridyl. This approach has been used in previous publications to determine the iron-dependency of DtxR repression in similar reporter systems (Ding et al., 1996). Using this approach, we observed a moderate but significant increase in Β-galactosidase activity, which would be expected if the deprivation of iron inactivated YtgR DNA-binding and alleviated repression of lacZ.

We have optimized a targeted chromatin immunoprecipitation qPCR assay to demonstrate direct and operator-dependent interactions between YtgR and the *trpBA* promoter element in the two-plasmid reporter system. This experiment revealed that YtgR specifically interacts with the *trpBA* promoter element, but not the promoter of trpR or dnaB, and that this interaction depends on the native operator sequence within the *trpBA* promoter. These studies were attempted in *Chlamydia*-infected cells with a polyclonal antibody raised against YtgC, but we were unable to efficiently immunoprecipitate cross-linked YtgR-DNA complexes with the raised antibody.

We were unable to conduct the biolayer interferometry (BLI) assay that we used in our previous publication because those experiments were performed by a collaborator (Dr. Scott Grieshaber) using the core facilities at the University of Florida at Gainesville. We do not have access to the BLI equipment anymore because Dr. Grieshaber has left Florida. Instead we decided to take advantage of our two-plasmid assay to demonstrate transcriptional repression of the lacZ reporter, and binding of YtgR to the IGR operator sequence via chromatin immunoprecipitation. While we were successful in establishing iron-dependent transcriptional repression by YtgR, we could not achieve consistent results with the controls and “unknown” samples in our ChIP assay in the presence of bipyridyl. We were able to pinpoint the issue with the addition of bipyridyl because other experiments we conducted, including the effects of the mutated operator sequence on DNA binding of YtgR, gave the expected outcomes with the control samples. However, we were unable to resolve the issue. The addition of bipyridyl relieved transcriptional repression in the doubly transformed *E. coli*, but its presence affected isolation of YtgR-DNA fragment interaction.

Despite these technical issues, we believe that our data collectively point to YtgR being an iron-dependent transcriptional repressor that targets the operator sequence within the conserved intergenic region, in conjunction with our previously published data that demonstrated YtgR binding to a different operator sequence in an iron-dependent manner.

Note that we have replaced the previous pCCT-lpdA negative control with that of pCCT-dnaB for consistency, i.e. other figures used the dnaB as negative control (Figure 2D).

2) The section on iron limitation and transcription read-through was not well written. It was hard to understand and not fully explained.

We have rewritten the section on transcription readthrough to enhance clarity.

The three graphs should be shown at the same scale so that they can be fairly compared.

We have revised the figure for clarity (Figure 7A-7B).

What is the significance of decreased read-through at 18 hpi compared to 12 hpi, and which should BpdI treatment be compared to? This is not a trivial consideration because the difference between 12 and 18 hpi could potentially be due to increased attenuation by tryptophan and/or iron levels.

We have re-analyzed our RT-qPCR data on transcription readthrough. To simplify the analysis, we are now only comparing two amplicons: one constant upstream “normalization” and various downstream “readthrough” amplicons. They include amplicons located upstream of the putative YtgR-dependent termination site, one that is located downstream of this site, and one at the predicted termination immediately after the TrpA CDS. RT-qPCR was performed for each amplicon separately for each mRNA sample. To further account for possible variation, the normalization amplicon was also included. Because this amplicon was common to all transcripts driven from the major trp promoter, it was used for normalization. This analysis is comparable to the analysis we performed in Figure 4B. In this instance, however, the collection of amplicons used to determine readthrough at the YtgR operator site excluded any that would detect transcripts originating from the alternative transcriptional start site, i.e. downstream of the *trpBA* promoter. We have also included a control amplicon as before at the 3’ end of the *trpA* sequence to indicate transcripts originating from the major trp promoter, AND those originating from the *trpBA* promoter, which when combined results in a ratio of >1.

The previous figure was not as intuitive as we would have liked, but to answer the reviewer’s question in reference to the previous figure, we can only speculate that the decreased readthrough at 18 h relative to 12 h could have been due to differences in iron availability that Chlamydia experienced during its development. Having said this, we reanalyzed the data using a different normalization amplicon. By doing so, there was no longer a difference between 12 and 18 h. This lack of difference was consistent across all amplicons monitored. We hope that this reanalysis addressed the reviewer’s additional concerns.

What does it mean that there was less read-through with depletion or iron than tryptophan? The presence of residual read-through despite iron depletion should be acknowledged and discussed.

This could be due to differences in promoter strength or robustness of transcription from the major trp promoter and the alternative promoter within the IGR that would translate differences in levels of transcripts. Assuming different promoter strengths, under Trp-starved condition, transcription initiates robustly from the major trp promoter, while under iron-limiting condition, transcription starts from the relatively weaker alternative promoter. The former would be expected to yield a higher ratio relative to the latter. This would be consistent with the data shown in Figure 7B where the readthrough ratio approaches 1 for the Trp-starved samples, indicating readthrough to complete transcription.

The result with tryptophan depletion was described as being expected but the reasoning was not given.

The results (Figures 3A and 6B) were expected based on previous reports by Carlson et al., Akers and Tan, Woods et al. that have already demonstrated the transcriptional induction of the trp operon under Trp depletion.

How do you know if tryptophan and iron affect read-through through the same or different mechanisms?

Regarding the mechanism of readthrough under iron- or Trp-depleted conditions, our argument would be that the mechanism is different. YtgR DNA-binding mediates readthrough under iron-limited conditions as a function of transcript termination, and inactivation of TrpR repression in the absence of the co-repressor Trp mediates readthrough under Trp-depleted conditions by inducing transcription initiation. However, the mechanism of Trp-dependent readthrough is only a side-note, with YtgR modulation of transcription termination being one of the major findings. Trp-depletion was simply used as a control because of its established regulation of trp operon transcription.

3) It's critical for the study to establish iron-starvation conditions that don't induce persistence, but 1 of the 2 persistence markers did not respond as expected. Additional transcriptional markers of persistence (Belland, 2003) should be tested to confirm that 6 hours of BpdI treatment did not induce persistence. Ideally these persistence marker should include iron-dependent and iron-independent genes.

We assume that the reviewer is referring to Figure 1. The purpose of this experiment was to identify a treatment duration that did not result in reduced replication (e.g. genome copy number), and aberrant expression of developmentally regulated genes (e.g. *euo* and *omcB*). We essentially discounted the time point where one or more of these criteria indicated persistence, and focused on the 6-h treatment for the rest of the manuscript. Rather than focus on the unexpected behavior of *omcB* at the “12+12” timepoint, we highlight the identification of a treatment protocol (e.g. “12+6”) that maintained normal developmental expression of *omcB* and *euo*. As further support for the “12+6” treatment condition, we (Brinkworth et al., 2018) published a genomewide analysis of the *C. trachomatis* transcriptome, and observed only a small collection of genes differentially expressed during iron starvation. These genes included trpB and trpA.

4) The RT-qPCR results in Figure 1D showing an increase in transcript levels of the late gene omcB following iron depletion are at odds with prior work from the group using a C. trachomatis serovar D isolate (Thompson and Carabeo, 2011) which showed decreased omcB transcript levels. This is touched on by the authors, who point out that omcB levels and iron stress responses seem to vary across chlamydial species and C. trachomatis serovars. This creates a potential concern that trpBA iron-mediated regulation might be LGV specific and may not occur in serovar D-K isolates which are more prevalent causes of infection in females (and the female reproductive tract in particular). This becomes problematic in relation to the hypothesis that YtgR-mediated regulation allows for differential responses to environments in the lower versus upper genital tract if serovars D-K do not show the LGV-type response.

We have edited the manuscript to discuss the inconsistency of *omcB* expression between this study and previous findings (subsection “Brief iron limitation via 2,2-bipyridyl treatment yields iron-starved, but non-persistent Chlamydia trachomatis”, first paragraph), the latter typically involved treatments at the start of infection. Not only has *omcB* expression been observed to vary under different stress conditions, but our decision to begin treatment at midcycle of chlamydial development necessarily means that previous reports that began stress treatment at the time of infection do not apply to this work and are not predictive for our studies. To discount the potential confounding effects of variations in omcB expression, we constrained our experiments to the 6 h treatment starting at 12 h post-infection.

Alignment of the IGS regions across serovars might address this point. Experimentally, the βgal assay (Figure 5) using non-LGV IGS regions and confirmation of iron regulated control of trpBA (RT-qPCR, Figure 3) in non-LGV serovars would support that the results are not LGV-specific and occur in serovars more frequently isolated from infected women. This would increase the significance of the study.

To address the concern about LGV vs. urogenital serovars, we have provided a sequence alignment of the *trpRBA* IGR between serovars L2, D, A and B which demonstrates a >99% sequence identity, with 100% conservation at the putative YtgR operator site (Figure 5—figure supplement 1, subsection “YtgR specifically binds to the *trpRBA* intergenic region in an operator-dependent manner to repress transcription of *trpBA*”, third paragraph). We had also noted this in the text in the initial submission.

5) While it is clear that trpR transcription terminates in the IGR, the data for iron-regulated termination within the IGR are fairly weak. For one, why were the comparisons in Figure 7C-E made to the 12hr instead of the 18hr time point, the latter of which appears to indicate a decrease in the ratio of both trpR and the IGR transcripts compared to read-through transcripts.

As noted above in our third response to point 2, we have re-analyzed the transcription readthrough data, which we believe addresses several of the concerns mentioned (Figure 7B, subsection “YtgR mediates iron-dependent termination of upstream transcripts at the putative *trpRBA* operator site”). We refer the reviewer to our third response to point 2 for details.

Secondly, Figure 3C shows increased trpR levels in iron depleted condition when compared to mock treatment at the same infection time point, similar to the iron-regulated genes shown in Figure 2, indicating this transcript is regulated by iron. Without more in-depth analysis of trpR promoter activity and trpR transcript stability, I'm not certain these results can be interpreted as iron-regulation of transcriptional read through.

We have two pieces of evidence that the trpR promoter is not subject to regulation by YtgR. First, as shown in Figure 5C that the presence of trpR promoter sequence did not confer YtgR-dependent transcriptional repression of the lacZ reporter. Second, the trpR promoter was not recognized by YtgR in the ChIP assay shown in Figure 5E. However, we cannot discount the possibility of a YtgR-independent iron-dependent regulation of the trpR promoter. What we would like to emphasize is that despite an iron-dependent, but YtgR-independent mechanism of transcriptional induction of trpR, presumably at the major trp promoter, the conclusion remains the same with regards to the role of YtgR in regulating trpR transcript termination. Indeed, in our β-galactosidase assay, the presence of DNA encompassing the trpR ORF and the intergenic region (IGR) in our *E. coli* heterologous system led to the repression of lacZ activity when YtgR was expressed. Both lacZ transcript and β-galactosidase enzyme activity were monitored. This indicated that YtgR binding to the IGR was sufficient to block RNA polymerase passage. To reiterate, the conclusion that YtgR mediates termination of transcription originating at the major trp promoter still stands, despite a possible iron-dependent, but YtgR-independent transcription regulation at the trp promoter. We also discuss the possibility that YtgR may function to attenuate transcription, and thus expression of trpR, which would offer a possible mechanism for the elevated expression of trpR observed by RT-qPCR under iron-limited conditions. Modifications to the text have been made to reflect the marginal iron-dependent regulation of trpR, and reinforce the conclusion that YtgR is sufficient to mediate termination of transcription of the trpR ORF.

6) The impact of the study would be increased by data supporting the model that YtgR binding to its operator decreases RNA polymerase processivity. While these studies would be challenging in Chlamydia especially in a 2-3 month time frame expected for the revision, they could be done with the heterologous transcription assay. This approach would allow the effect of YtgR and other transcription factors on read-through transcription to be compared. It could also be used to show if this effect is dependent on iron or tryptophan. One concern regarding iron-regulated termination is that the data are not wholly consistent with this idea. Specifically, the data show some modest iron regulation of trpR, depending on what time points are used to evaluate the data (see Figures 3 and 7). Moreover, there are two TTS in the IGR downstream of the YtgR operator, suggesting termination can occur independently of YtgR binding. Thus it is possible that multiple factors, including RNA stability, RNA processing, and termination, are contributing to their observations. In the absence of additional data, the authors should temper their conclusions.

To address the above concern about RNA polymerase processivity, we have adapted our two-plasmid system such that the reporter plasmid contained the entire *trpR* ORF and *trpRBA* IGR upstream of the *lacZ* reporter gene (Figure 7C). Thus, RNAP initiated at the pBAD element would be required to readthrough the whole endogenous *trpR*-IGR sequence prior to reaching *lacZ*. We then monitored readthrough by RT-qPCR (as explained above) in the presence or absence of YtgR (Figure 7D). Using this system, we observed a significant decrease in the amount of readthrough to the *lacZ* gene when YtgR was expressed, consistent with the idea that YtgR binding to the IGR inhibits RNAP processivity and induces transcript termination. Moreover, we were able to use this system to demonstrate a significant reduction in β-galactosidase activity when YtgR is ectopically expressed (Figure 7E), confirming that *lacZ* is indeed not being transcribed similarly under these conditions. We have commented on the fact that we cannot discount a possible role for the other TTSs in this iron-dependent regulation (Discussion, third paragraph), but we argue that any iron-dependent regulation of the other TTSs would produce the same result of limiting *trpBA* expression. Nevertheless, we have demonstrated that there is iron-dependent termination at the YtgR operator and that YtgR regulation of the IGR inhibits readthrough. The most parsimonious model is therefore that YtgR binding to its operator site in the IGR terminates transcription from the upstream *trpR* promoter, limiting downstream *trpBA* expression.

7) Is this the first demonstration of iron regulating a C. trachomatis virulence trait?

With obligate intracellular pathogens like Chlamydia, it is difficult to make distinctions between virulence and survival. However, factors that are considered virulence factors in other pathogens have been demonstrated in Chlamydia to respond to iron starvation. For example, a report from Slepenkin et al., 2007 described the induction of transcription of various components of the type III secretion system upon treatment with the iron chelator Desferal. We found a similar transcriptional induction of components of the type III secretion system in shorter iron starvation by bipyridyl (Brinkworth et al., 2018).

This should be put into broader context of bacterial pathogenesis, where iron regulation of virulence is a recurrent theme for many pathogens, and highlighted as a new twist on this theme.

We have commented on two aspects of chlamydial virulence (or pathogenesis) in the female genital tract that are likely impacted by iron-dependent YtgR regulation of *trpBA*: the possibility of deploying a unified response to counter the pleiotropic insults of IFN-γ stimulation (Introduction, last two paragraphs), as well as the possibility that iron-dependent regulation of *trpBA* may have a specific and unique role in activating tryptophan biosynthesis in the lower genital tract, where the substrate for salvage, indole, is available, allowing *C. trachomatis* to prepare for oncoming IFN-γ stress (Discussion, fifth and sixth paragraphs). We have added an additional statement that this may play an important role in virulence in genital serovars of *C. trachomatis*.